# Tensor-Parallelism with Partially Synchronized Activations

**Itay Lamprecht**[†‡]  **Asaf Karnieli**[†]  **Yair Hanani**[†]  **Niv Giladi**[○*]  **Daniel Soudry**[‡]

[†]Intel, Israel
[‡]Department of Electrical and Computer Engineering - Technion, Haifa, Israel
[○]AWS AI Labs

```
{itay.lamprecht, yair.hanani}@intel.com
           {ngiladi}@amazon.com
  {asafkarnieli, daniel.soudry}@gmail.com
```

## Abstract

Training and inference of Large Language Models (LLMs) with tensor-parallelism requires substantial communication to synchronize activations. Our findings suggest that with a few minor adjustments to current practices, LLMs can be trained without fully synchronizing activations, reducing bandwidth demands. We name this "Communication-Aware Architecture for Tensor-parallelism" (CAAT-Net). We train a 7B parameter CAAT-Net model and show that tensor-parallel communication can be reduced by up to 50% with no significant drop in pretraining accuracy across nearly all evaluated benchmarks. We also experiment with smaller 130M and 1.1B models to show the robustness and scalability of our method. We find that, in some scenarios, validation loss can even improve when reducing communication. Finally, we demonstrate how CAAT-Net accelerates both training and inference workloads across various settings and model sizes.

## 1 Introduction

As Large Language Model (LLM) training continues to scale, often requiring the use of hundreds or even thousands of devices, the need for efficient distributed training and inference techniques is constantly growing. Tensor-parallelism [1] is a critical component in the training of many well-known LLMs, such as GPT-3 [2], Llama [3, 4], and BLOOM [5], enabling the efficient utilization of hardware resources to support models with billions of parameters. By partitioning weight tensors across multiple devices, tensor-parallelism allows each device to store only a fraction of the model's weights during computation. This significantly reduces the memory footprint per device, making it feasible to train extremely large models on hardware with limited memory capacity. The efficiency of tensor-parallelism in reducing training memory has made it a cornerstone technique for distributed training. During inference, where the LLM response time is critical, distributing the computation over multiple devices using tensor-parallelism can significantly reduce latency.

Much of the total time spent training or using LLMs is consumed by the communication overhead inherent in tensor-parallelism, which arises from the need to synchronize activation or gradient tensors. Specifically, synchronization is done by all-reduce operations, where tensors from all devices are summed into a single tensor, which is copied to all devices. This operation occurs multiple times in each forward and backward pass and constitutes a substantial fraction of the total workload time.

A recent approach to addressing this issue is pipelining communication and computation [6] in such a way that the communication is overlapped with the computation and the footprint of communication is reduced. However, this approach is limited: when tensor-parallelism is expanded to use more

---

[*]This work was done while the author was at Intel.

devices, the compute workload per device decreases, while communication payload per device remains relatively constant [7]. This means that the relative cost of communication grows as the compute-to-communication ratio decreases. In extreme cases, communication time can overcome computation time, and thus dominate the training process. For these reasons, even the largest language models are typically trained with a tensor-parallelism dimension of 8 [4], utilizing fast intra-node communication for the heavy all-reduce operations.

Improving tensor-parallelism efficiency is even more important given that the growth in compute power exceeds the growth in communication bandwidth [7]. This trend should further expose communication time in large-scale training. Minimizing tensor-parallelism communication can enable better hardware compute utilization and reduce overall training costs, especially when extending tensor-parallelism across nodes. This is in line with the recent trend of building multi-node systems with high bandwidth communication.

In traditional tensor-parallelism, the activation tensors after communication are identical on all devices, i.e., fully synchronized. In this work, we show that LLM training can converge without fully synchronizing the activation tensors in the all-reduce operation. This means that we allow activations to vary on different devices after communication. We show that without full synchronization, the current training practice needs to be slightly adjusted. Failing to do so leads to critical issues such as a mismatch between forward and backward passes and numerical issues, which often result in training divergence. Relying on this insight, we suggest the partial channel–reduce operation, in which only a subset of the channels in the hidden dimension of the activation tensors is reduced. Unlike regular all-reduce, activations are not identical on all devices after the partial channel–reduce operation. In the extreme case where no channels are synchronized in partial channel–reduce, the model resembles an ensemble, communicating only to compute the loss function and embeddings. In the case where all channels are reduced, the model is a vanilla transformer model. We introduce Communication-Aware Architecture for Tensor-parallelism (CAAT-Net) — a new model architecture that is tailored for tensor-parallelism by utilizing partial channel–reduce to decrease communication overhead. While CAAT-Net has a smaller communication overhead compared to an identical model with full all-reduce, the number of parameters and total compute stay the same.

We train a Llama2-7B model [8] with partial channel–reduce over 160B tokens and show that there is no significant degradation in nearly all evaluation benchmarks we tested, while reducing the communication payload by 50%. Furthermore, we train multiple variants of the 1.1B parameter TinyLlama model [9] and a smaller 130M parameter model. We study the effects of the number of synchronized channels and tensor-parallel dimension on accuracy. We find that a gradually reducing communication from full synchronization first yields a slight improvement in validation loss, but performance worsens when communication becomes too limited. Reducing the communication by 50% achieves either similar or slightly better validation loss for all models we tested. Finally, we show the training and inference speedup of our proposed method in various settings.

In summary, our contributions in this paper are as follows:

- We show that when using tensor-parallelism, LLMs can be trained without fully synchronizing activation tensors.
- We propose CAAT-Net, a novel architecture that significantly decreases communication traffic in training with tensor-parallelism by synchronizing only part of the activation tensors.
- We show that in various settings, CAAT-Net accelerates both training and inference, and achieves accuracy largely on par with fully synchronized training.

## 2   Related Work

The challenge of efficient training and inference on a large scale has attracted much attention, both in the engineering and research fronts, in close correspondence with each other. While most of the literature focuses on reducing data-parallel communication, there have been a few works that focus on tensor-parallelism. Our approach is orthogonal to all methods covered in this section.

**Pipelining computation and communication.** Some methods accelerate training with tensor-parallelism by overlapping communication and computation, [6, 10–12]. In Domino DeepSpeed, the training batch is split into smaller pieces, and data dependency is broken such that communication is not on the critical path and can be overlapped with computation. Alternatively, Ladder Residual

pipelines communication and computation by changing the transformer architecture, such that each layer uses an earlier version of the input from two layers back. This allows it to compute while the previous layer's results are still being communicated. While these approaches achieve speedup in inference and training, they are limited — there are many cases in which the compute time is not sufficient to fully hide tensor-parallel communication. For example, with a growing tensor-parallel dimension, the computation per device is reduced while the communication is not, so overlapping efficiency is reduced as well. Furthermore, other parallelism types interfere with tensor-parallel related communication. One of many examples is context-parallelism, where communication time alone can exceed compute time [13]. When adding tensor-parallelism after attention blocks that use context-parallelism, there is no compute left to pipeline both types of communication. For this reason we view our method, which reduces the total tensor-parallel communication, as complementary to pipelining methods.

Additionally, in NVIDIA GPUs such as H100/H200, pipelining communication and computation comes with a cost — the Stream Multiprocessors (SM), which perform the computation, are also needed for communication. When pipelining communication and computation, some SM cycles are spent on communication-related tasks instead of computation. Furthermore, while pipelining computation and communication can accelerate training and inference, our method reduces total communication, and can potentially improve other aspects such as power consumption.

**Compressed communication.** Another approach is to address the communication overhead by compressing the communication itself at the tensor-parallel or pipeline-parallel stages [14, 15]. This can be done using various methods, such as auto-encoders or sending the Top-K elements of the tensors. While compression succeeded in other aspects of optimization, it has yet been demonstrated that compression works for tensor-parallelism, to the best of our knowledge. In fact, we observed severe degradation in model accuracy with even minimal compression using Top-K or random masking to compress activations (our observations can be found in Section 5.3). Additionally, compression introduces additional computation overhead which can reduce the speedup achieved by reducing communication. Our method differs from these methods because it does not compress activations. Although not all activations are shared between devices, those that are not shared are still used.

**Asynchronous optimization.** A different approach, mainly focused on mitigating overhead introduced by data-parallel communication, introduces asynchronism to the optimization by not waiting for all communication or straggling devices. In this case, different devices store all of the model weights, but each device can hold a slightly different copy. Asynchronous training is inherently more scalable than synchronous training by being robust to all kinds of worker and communication faults. However, this scalability comes with a price: convergence difficulties and generalization deterioration [16]. In contrast, when training with tensor-parallelism, each device stores a different subset of the model weights. Therefore, while we do not completely synchronize activations, we do not encounter the problem of multiple workers storing a different copy of the same weight, and training remains synchronous.

**Post-training techniques.** There have been many attempts to reduce tensor-parallel communication using post-training techniques, such as quantization [17, 18], and selectively removing complete communication points after the attention operation [19]. While these techniques are promising, they are relevant only for decreasing the communication bottleneck during inference.

## 3 CAAT-Net

To reduce communication bandwidth in LLM training and inference, we replace the all-reduce operation, designed to synchronize activations between devices, with a partial channel–reduce operation, such that only a subset of the channels in the hidden dimension is synchronized between devices. In partial channel–reduce, channels that are sent to other devices are referred to as shared channels, while those that are unique per device are private channels. The shared channels are chosen at initialization to be the first $h \cdot p$ channels, where $h$ is the hidden dimension size and $p$ is a synchronization factor controlling how much of the hidden dimension is synchronized. For $p = 1$ the partial channel–reduce turns into a full all-reduce operation (no private channels). The operation is visualized in Figure 2 (c). In this section we present a novel communication-aware model architecture which utilizes partial channel–reduce to accelerate LLM training and inference. A theoretical speedup analysis of our method is available in Appendix C.

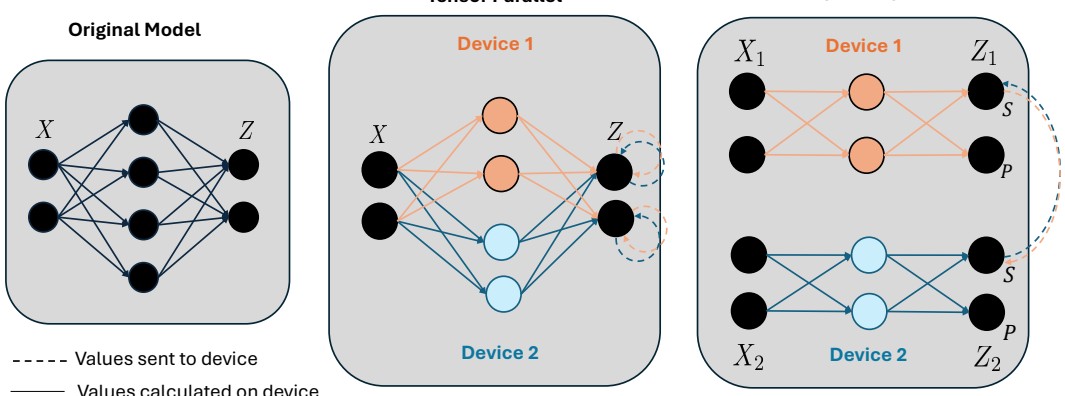

Figure 1: **CAAT-Net model architecture. Left.** We exemplify our approach on a two-layer fully-connected neural network on a single device. **Middle**. When using tensor-parallelism with two devices, the input activation $X$ is identical on both devices. Each device uses its own set of weights to multiply the inputs, yielding intermediate activations, which are then reduced into identical copies of $Z$ on both devices. **Right.** CAAT-Net receives different input activations, $X_1$ and $X_2$, for Device 1 and Device 2, respectively. These yield intermediate activations that are partially synced between the devices, producing $Z_1$ and $Z_2$ on Device 1 and Device 2, respectively. Private channels are marked $P$, and shared channels are marked $S$.

## 3.1 Architecture

While changing the communication primitive from all-reduce to partial channel–reduce seems like a minor change, it effectively changes the model architecture. Partial channel–reduce fundamentally alters the model's computation graph and information flow. This distinguishes CAAT-Net from communication optimizations like pipelining or compression, which preserve or approximate the underlying mathematical operations. Consequentially, the Multi Layer Perceptron (MLP) and attention layers must be defined differently when taking partial channel–reduction into account. In this section we define the MLP using partial channel–reduce. The definition of the attention operation is described in Appendix A.1.

**Standard all-reduce in MLP.** Before mathematically describing the MLP using partial channel–reduce, we describe the MLP using regular all-reduce in a tensor-parallel setting. For simplicity, we examine a case where the tensor-parallel dimension is 2. For the general case, refer to Appendix A.2. Given consecutive MLP weight matrices $A$ and $B$, input tensor $X$ and an activation function $\sigma$, the output of the MLP layer $Z$ is given by

$$Y = \sigma(XA) \quad ; \quad Z = YB \tag{1}$$

where $Y$ is the output of the first MLP layer. The basic architecture of a transformer MLP is visualized in Figure 1 (left). When using tensor-parallelism, we partition $A$ along its columns, i.e.,

$$A = [A_1, \ A_2] \tag{2}$$

so the computation of $Y$, is

$$Y_1 = \sigma\left(XA_1\right) \quad ; \quad Y_2 = \sigma\left(XA_2\right) \quad ; \quad Y = [Y_1, \ Y_2] \ . \tag{3}$$

We partition $B$ along its rows, i.e.

$$B = \begin{bmatrix} B_1 \\ B_2 \end{bmatrix} \tag{4}$$

and, after an all-reduce, the output of the MLP is

$$Z = Y_1 B_1 + Y_2 B_2 \ , \tag{5}$$

where $Y_1 B_1$ is computed on the first device and $Y_2 B_2$ is computed on the second. The implementation of all-reduce with tensor-parallelism is visualized in Figure 1 (middle).

**Partial channel–reduce in MLP.** We first note that with partial channel–reduce, the input to the MLP will be different per device (as the previous attention layer will also use partial channel–reduce in its output). Therefore, the inputs to each device are denoted $X_1$ and $X_2$, as is shown in Figure 1 (right). Thus, the output of the first linear layer when using partial channel–reduce is:

$$Y_1 = \sigma\left(X_1 A_1\right) \quad ; \quad Y_2 = \sigma\left(X_2 A_2\right) \quad ; \quad Y = [Y_1, \, Y_2] \, . \tag{6}$$

Next, we need to partition $B$ along both its columns and rows

$$B = \begin{bmatrix} B_{11} \ B_{21} \\ B_{12} \ B_{22} \end{bmatrix} . \tag{7}$$

Then, the outputs of the MLP with partial channel–reduce, denoted $Z_1$ and $Z_2$, are different per device and are calculated by

$$Z_1 = [Y_1 B_{11} + Y_2 B_{12}, \, Y_1 B_{21}] \quad ; \quad Z_2 = [Y_1 B_{11} + Y_2 B_{12}, \, Y_2 B_{22}], \tag{8}$$

where $B_{11}$ and $B_{21}$ are on the first device, and $B_{12}$ and $B_{22}$ are on the second device. The values in $Y_1 B_{11} + Y_2 B_{12}$ are in the shared channels, and $Y_1 B_{21}$ and $Y_2 B_{22}$ are both in the private channels, which are visualized in Figures 1 and 2.

### 3.2   CAAT-Net Inference

Models trained with CAAT-Net use partial channel–reduce in inference as well. This leads to communication reduction and speedup when serving models, as we show in Section 5.2.

If the model is served with the same tensor-parallel dimension that it is trained with, serving the model is straightforward. This is different when using the model in inference with tensor-parallel dimension different than in training. As an example, we examine a scenario where the model is trained with a tensor-parallel dimension of 2, and inference is performed with a single device. In training, the output of a transformer layer is different on each device. In inference, one device needs to handle both of these copies. This can be done using 'logical devices'. In this case, a single physical device can simulate multiple tensor-parallel ranks, and sequentially calculate the output of each layer for every 'logical' tensor-parallel device. Furthermore, the partial channel–reduce is replaced with local summation inside the single device. It is also possible to return to a full all-reduce operation during inference through fine tuning, to increase the value of $p$ to 1. Then, a regular transformer architecture is achieved, and inference can be run on any tensor-parallel dimension.

## 4   Partial Synchronization: Implementation

To train LLMs without full synchronization of activation tensors, there are two main adjustments that must be done to current training frameworks [1]. The first is an adjustment to the backward implementation of tensor-parallelism, which is necessary to avoid forward-backward mismatch in training. The second is accumulating tensors in 32 bit (fp32) in the all-reduce of the backward pass. We find empirically that this is necessary to avoid numerical issues in training.

**Forward-backward mismatch.** To train LLMs with partial activation synchronization, we need to further examine the backward pass of a transformer model. In traditional LLM training, the all-reduce operation in the backward pass is done on the neural gradients produced by an MLP or attention layer, as shown in Figure 2a. Explicitly calculating the gradients propagated through the network, we find that the all-reduce operation can be applied in multiple locations in the backward pass without altering the backpropagation algorithm. We also find that in the case of partially synchronized activations, the reduction operation can only be applied in one place. To show this analytically we examine the input to an MLP layer on device $m$, denoted $X_m$, as a function of the output of a previous attention layer on device $m$ before the all-reduce operation, denoted $Z_m$, with $R$ being the residual connection, we have

$$X_m = \text{Norm}\left(\sum_m Z_m + R\right), \tag{9}$$

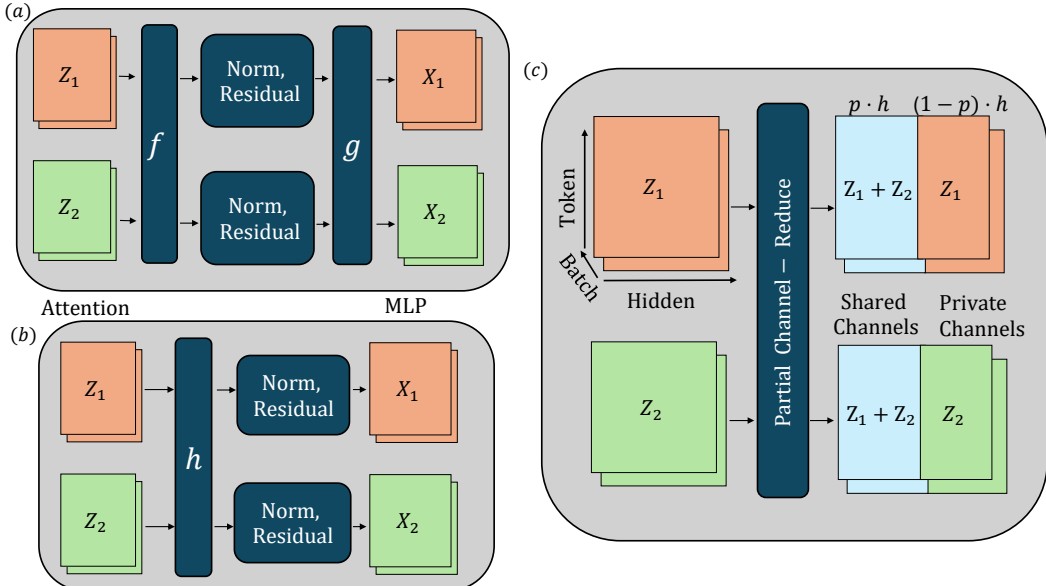

Figure 2: **Partial synchronization and partial channel–reduce. (a)** Vanilla transformers in current training frameworks. The operation $f$ is an all-reduce in the forward pass and identity in the backward pass. The operation $g$ is an all-reduce in the backward pass and identity in the forward pass. **(b)** With partial synchronization, $h$ denotes the reduction operation in both the forward and the backward pass, since both must be done at the same location. Synchronization of the normalization function parameters is necessary. **(c)** Partial channel–reduce with parameter $p$ over 2 devices.

where Norm can be any normalization function (typically LayerNorm [20] or RMSNorm [21]). We examine the loss $\mathcal{L}$ as a function of $X_m$ for all $m$, and $X_m$ as a function of $Z_1$, and calculate the derivative of the loss function w.r.t $Z_1$. For simplicity, we ignore below the residual connections (in Appendix B we show the derivation including the residual). Using the chain rule:

$$\frac{\partial \mathcal{L}\left(X_1\left(Z_1\right),..,X_M\left(Z_1\right)\right)}{\partial Z_1} = \sum_m \left(\frac{\partial \mathcal{L}\left(X_m\right)}{\partial X_m} \cdot \frac{\partial X_m\left(Z_1\right)}{\partial Z_1}\right) = \sum_m \left(\frac{\partial \mathcal{L}\left(X_m\right)}{\partial X_m} \cdot \frac{\partial X}{\partial Z_1}\right) \cdot \quad (10)$$

In the second equation, we recalled that $\frac{\partial X_m}{\partial Z_1}$ are equal for all $m$ (denoted now as $\frac{\partial X}{\partial Z_1}$) because in tensor-parallelism with full all-reduce operations $X_1\left(Z_1\right) = X_2\left(Z_1\right) = ... = X_M\left(Z_1\right)$, and thus their derivative w.r.t. $Z_1$ are also equal. Here, the summation over $m$ is the all-reduce operation in the backward pass. Our key insight here is that, since the matrix multiplication distributes over addition, i.e.

$$\sum_m \left(\frac{\partial \mathcal{L}(X_m)}{\partial X_m} \cdot \frac{\partial X}{\partial Z_1}\right) = \sum_m \left(\frac{\partial \mathcal{L}(X_m)}{\partial X_m}\right) \cdot \frac{\partial X}{\partial Z_1}, \quad (11)$$

the all-reduce operation can be used before back-propagating through the normalization function (right side of the equation), or after the normalization function (left side of the equation). In standard training frameworks, such as Megatron-LM, the all-reduce operation is used before back-propagation through the normalization function (i.e., the 'g' operation in Figure 2a). When training with partial synchronization, the assumption that $\frac{\partial X_m}{\partial Z_m}$ are equal for all $m$ no longer holds. For this reason, the all-reduce operation in the backward pass must occur after the calculation of the normalization function derivative (i.e., the 'h' operation in Figure 2b) so it would match the location of the summation in the forward pass (i.e., the 'f' operation in Figure 2a), to get the correct gradients.

Similarly, we examine the normalization function parameter update, denoted $\beta$, with a full all-reduce operation:

$$\frac{\partial \mathcal{L}\left(X_1\left(\beta\right),...,X_M\left(\beta\right)\right)}{\partial \beta} = \sum_m \left(\frac{\partial \mathcal{L}\left(X_m\right)}{\partial X_m} \cdot \frac{\partial X_m}{\partial \beta}\right) \cdot \quad (12)$$

In the gradient descent step, after each iteration, before updating the weights, the updates $\frac{\partial \mathcal{L}}{\partial \beta}$ need to be synchronized with an all-reduce operation. In popular training frameworks, the fact that $\frac{\partial X_m}{\partial \beta}$ is

identical for all $m$ devices is used once again. There is no need for gradient synchronization after each step, if the all-reduce happens before backpropagating through the normalization function:

$$\sum_m \left( \frac{\partial \mathcal{L}(X_m)}{\partial X_m} \right) \cdot \frac{\partial X}{\partial \beta}, \tag{13}$$

where, again, $\frac{\partial X_m}{\partial \beta}$ are equal for all $m$ (denoted as $\frac{\partial X}{\partial \beta}$). To summarize, when using full all-reduce, the following are equivalent:

- Reduce the neural gradients before they backpropagate through the normalization function.
- Reduce the gradients after they backpropagate through the normalization function, and reduce normalization function parameters before the weight update.

However, when training with partial synchronization, such as partial channel–reduce, only the second approach is possible. Our altered backward implementation is detailed in Figure 2b. It is important to note that the number of normalization function parameters per layer is typically $\Theta(h)$, where $h$ is the hidden size, and the synchronization of normalization function parameters happens after every step. For these reasons, the additional communication of normalization function parameters is negligible in comparison to the tensor-parallel all-reduce operation.

**Numerical stability via 32 bit gradient accumulation.** In the previous section we presented two different implementations of the backward pass. If we fully synchronize activations in the forward pass, these implementations are mathematically identical. Despite this equivalence, we found in our experiments that there can still be a large gap in training convergence when using our alternative implementation, even when activations are fully synchronized. This gap arises due to numerical differences between the implementations, and it can be completely mitigated by accumulating the reduced gradients in 32 bit precision instead of 16 bit precision. While LLMs are conventionally trained in 16 bit precision, there are some operations where 32 bit precision is critical, such as in the accumulation in matrix multiplication operations, or normalization layers. While the tensor-parallel gradient all-reduce is typically not one of these operations, we find that when moving the all-reduce in the backward pass, 32 bit accumulation is crucial. It is important to note that while accumulation needs to be done in 32 bit precision, the communication itself can be done in 16 bit precision, and values are simply upcast after communication and before accumulation. Loss curves comparing our training experiments with all-reduce accumulation in fp32 and bfloat16 are available in Appendix E.1.

**Private channel scaling.** Partial channel–reduce leads to differences in the statistics of shared and private channels, which affects signal propagation in the network. Consider the case of partial channel–reduce with 2 tensor-parallel devices. Assuming each element in the MLP outputs before reduction $(Y_1 B_{11}, Y_2 B_{12}, Y_1 B_{21})$ has zero mean and variance $\sigma_A^2$, and are independent between devices. The activation variance for shared channels is

$$\text{Var}(Y_1 B_{11} + Y_2 B_{12}) = \text{Var}(Y_1 B_{11}) + \text{Var}(Y_2 B_{12}) = 2\sigma_A^2, \tag{14}$$

and for private channels is:

$$\text{Var}(Y_1 B_{21}) = \sigma_A^2 \tag{15}$$

This variance mismatch can lead to uneven signals across channels in both the activation and gradient calculations. To correct this mismatch, we multiply the activations in the private channels by a corrective factor of $\sqrt{2}$ (or $\sqrt{r}$ if we train with a tensor-parallel dimension of $r$). This way, after the partial channel–reduce, the activations have identical variances over all channels. Ablation studies for private channel scaling are available at Appendix E.3. While we find a slight improvement in validation loss across values of $p$, private channel scaling is not mandatory to achieve sufficient accuracy results.

## 5 Experiments

Experiments* were conducted with Intel Gaudi3 HPU accelerators. Gaudi3 has 128GB on-board memory. Each device has 525 GB/s intra-node connection and 75 GB/s inter-node connection.

---

\* Code is available at https://github.com/itlamp/Megatron-LM-comms

Table 1: **CAAT-Net vs baseline: Zero-shot accuracy after pretraining.** 7B parameter models, with $p = 0.5$ and tensor-parallel 8.

| Model | LAMBADA (acc) | Hellaswag (acc) | WinoGrande (acc) | PIQA (acc) |
|---|---|---|---|---|
| Baseline | **61.34 ± 0.68** | 45.85 ± 0.50 | 61.48 ± 1.37 | **72.91 ± 1.06** |
| CAAT-Net | 61.05 ± 0.68 | **46.10 ± 0.50** | **62.19 ± 1.36** | 72.86 ± 1.04 |
| | OpenBookQA (acc) | BOOL-Q (acc) | WikiText (ppl) | Validation Loss |
| Baseline | **26.60 ± 1.98** | **64.89 ± 0.83** | 12.51 | 1.01 |
| CAAT-Net | 24.00 ± 1.87 | 62.51 ± 0.85 | **12.46** | **1.00** |

## 5.1 Large Scale Training

We trained a variation of Llama2-7b [8] with partial channel–reduce. Training was conducted from scratch over 160B tokens from the RedPajama dataset, spanning 8 nodes, each containing 8 accelerators. We chose a partial channel–reduce hyperparameter of $p = 0.5$ and a tensor-parallel dimension of 8, with all other hyperparameters identical to those used in training of the original model. We choose these values of $p$ and tensor-parallel dimension as an example which we expected, based on our ablation studies, to yield good accuracy results, while significantly reducing network bandwidth.

We evaluate the model's performance on a diverse set of common sense tasks selected from the Language Model Evaluation Harness framework [22]. The chosen benchmarks include tasks such as HellaSwag [23], WikiText103 [24], LAMBADA [25], WinoGrande [26], PIQA [27], OpenBookQA [28] and BoolQ [29]. The zero-shot results are available in Table 1. All accuracy results are not statistically significant, except for Bool-Q, which is marginally significant (p-value = 0.046 using Welch's t-test).

To further study the scalability and robustness of our method, we train 130M and 1.1B models in multiple settings. The 1.1B model used is TinyLlama [9], and is trained over 100B tokens. The 130M parameter model is a small custom model based on the LLaMA architecture. It has 16 attention heads and a hidden dimension size of 768. It was trained with an initial learning rate of $6 \cdot 10^{-4}$, with the AdamW optimizer. The training was performed with a global batch size of 256 and a sequence length of 1024. The architecture consists of 12 transformer layers with multi-head attention. It is trained over 7.8B tokens. Both models are trained on the RedPajama dataset, using the GPTSentencePiece tokenizer.

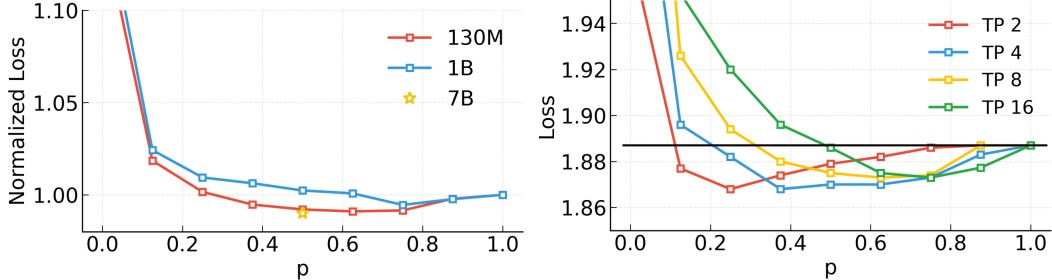

Figure 3: **Training accuracy in multiple scenarios. Left.** Validation loss of 130M and 1.1B models for different values of $p$, and of the 7B model with $p = 0.5$, normalized to the loss at $p = 1$. **Right.** Validation loss for the 130M model with varying values of $p$ and tensor-parallel dimension (TP).

We train both models with varying values of $p$, with tensor-parallel dimension of 8. Results are available in Figure 3 (left). We find that for large values of $p$ (over 0.75), the 130M and 1.1B models behave similarly. For intermediate values of $p$ (between 0.75 and 0.25), the smaller 130M model has a slight improvement in validation loss compared to the baseline, while the 1.1B model is close to or slightly above the baseline. For smaller values of $p$, there is degradation for both models. Furthermore, we added the 7B model we trained with $p = 0.5$, which shows a slight improvement w.r.t the baseline, showing potential in further scaling our method. Zero-shot accuracy for all 1.1B experiments is available in Appendix E.2.

We also study the effects of tensor-parallel dimension on the 130M model. Results are available in Figure 3 (right). For all tensor-parallel dimensions, gradually decreasing $p$ from 1 initially leads to a slight improvement in validation loss, but performance deteriorates once $p$ becomes too low. As the tensor-parallel dimension increases, this degradation begins at a higher value of $p$. For all tensor-parallel dimensions, $p = 0.5$ does not degrade or improves validation loss.

Finally, we train an additional model, other than the models from the Llama architecture, to show the robustness of our method. Specifically, we trained GPT3-XL [2] with tensor-parallel 8 and $p = 0.5$, on the RedPajama dataset using the GPTSentencePiece tokenizer, rotary positional embeddings and with a global batch size of 512. We trained for a total of 50B tokens. Similarly to results on Llama, we find that CAAT-Net maintains accuracy results. The full results are available in Appendix E.2.

## 5.2 Speedup

We examine Llama2-7b with partial channel–reduce to measure inference and training speedup. In this section we show relative speedup (i.e., speedup compared to the baseline). For absolute measurements, see Appendix E.5. Furthermore, for inference, we experiment with larger models (34B and 70B) and achieve similar results to those reported in this section. Full results for these models are available in Appendix E.4.

**Inference.** To measure inference speedup, we report the Time-to-First-Token (TTFT) for varying batch sizes, using a prompt length of 2K tokens. We use partial channel–reduce with tensor-parallel 8 and 16. We compare results against the baseline, which is Intel's Optimum-Habana without any of our adjustments. The results are shown in Figure 4 (bottom). For tensor-parallel 16 and $p = 0.25$ we measure a maximum speedup of 26% with batch size 32. In the large scale training setting of $p = 0.5$ and tensor-parallel 8, which we show does not degrade accuracy, we obtain a maximum speedup of 14% for batch size 32.

Furthermore, to show that accelerating workloads using CAAT-Net is not specific to Intel Gaudi hardware, we conduct experiments on NVIDIA hardware. Our experiments were done using the gpt-fast repository, and consisted of replacing all-reduce with partial channel–reduce during inference. We conducted experiments on 8 NVIDIA H100-80GB-HBM3, and 8 NVIDIA A100-SXM4-80GB, both with NVLink. Results are shown in Figure 4 (top right). We find that our method speeds up inference TTFT by up to 13% on NVIDIA hardware with $p = 0.25$ and tensor-parallel dimension 8. Speedup on Gaudi is more significant in our experiments, but we believe further software performance optimizations on NVIDIA hardware can bridge this gap. Full measurements, including for additional batch sizes, are available in Appendix E.6.

Alongside accuracy considerations, practitioners should consider hardware constraints when selecting $p$. For example, on Gaudi3, speedup results were more significant when selecting $p$ such that the communication volume is a multiple of the communication buffer size. Specifically, when experimenting with Llama2-7B with batch size 16, we identified that the optimal performance speedup is achieved with a value of $p$ which is a multiple of $2^{-7}$ (i.e., the number of channels reduced is a multiple of 16). As a result, selecting $p = 0.703125$, which is equal to $90 \cdot 2^{-7}$, achieves better speedup than $p = 0.7$.

**Training.** To measure training throughput speedup, we report the tokens per second (TPS), for tensor-parallel dimensions 4, 8, and 16, with varying values of $p$. Tensor-parallel 4 experiments were conducted with a micro-batch size of 4, while tensor-parallel 8 and 16 experiments were conducted with a micro-batch size of 8. The results are shown in Figure 4 (top left). We compare results against the baseline, which is Intel's Megatron-LM fork without any of our adjustments. For training in tensor-parallel 16, our method can improve throughput by up to 14% when selecting $p = 0.25$ and by 9% when selecting $p = 0.5$.

All experiments were conducted using private channel scaling, as described in Section 4. Disabling it provides an additional 1–2% speedup, with only minor accuracy degradation (see Appendix E.3), making training without private channel scaling a reasonable alternative. It is important to note that in training speedup experiments, for all values of $p$ and the baseline, accumulation in the backward pass was done in 16-bit precision due to technical implementation limitations. For this reason, the training throughput results in this section should be seen as a performance projection. We do not expect accumulation in fp32, if communication is kept in 16-bit precision, to significantly affect the speedup reported in this section.

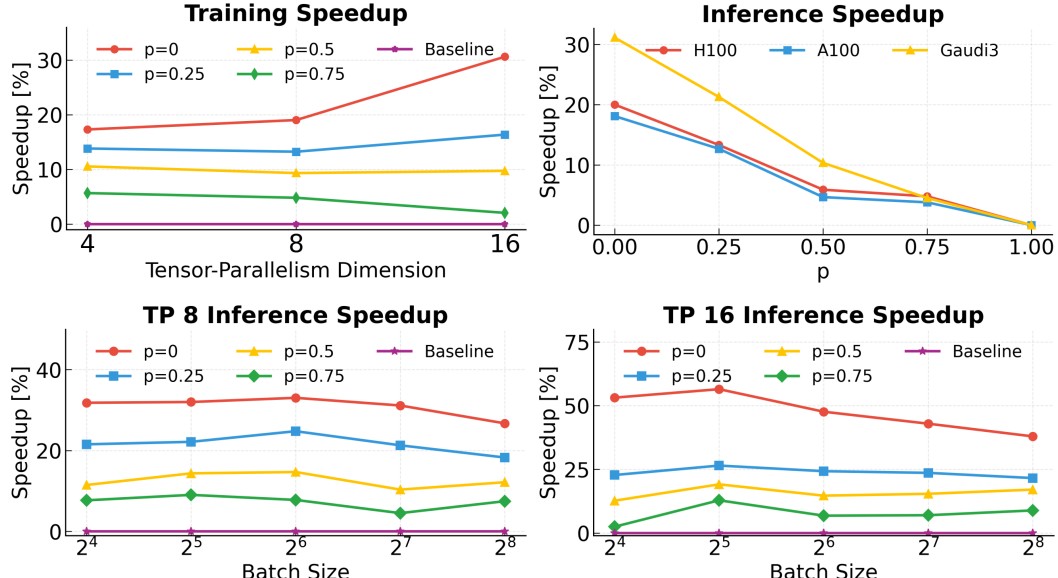

Figure 4: **Speedup in training and inference for a Llama 7B model. Top Left** Training speedup for varying values of $p$ and tensor-parallel dimension. **Top Right.** Inference Time-To-First-Token (TTFT) speedup on different hardware, using batch-size 128. **Bottom Left.** Inference TTFT speedup using tensor-parallel 8 as a function of batch size, for different values of $p$. **Bottom Right.** Inference TTFT speedup using tensor-parallel 16 as a function of batch size, for different values of $p$.

## 5.3 Comparison to compression methods

To compare our method to compression methods, we use two alternative approaches for compressing activation communication. While CAAT-Net preserves low validation loss, we report severe degradation for both compression methods, even with minimal compression. The first method is reducing communication using a random mask. The values that are multiplied by 0 in the random mask are not communicated. This is effectively applying dropout before communication. The second method is using a Top-K mask, which selects the Top-K entries in each token of the activation tensor and discards the others. In both cases, we apply the mask in the forward pass before communication. To avoid a forward-backward mismatch,

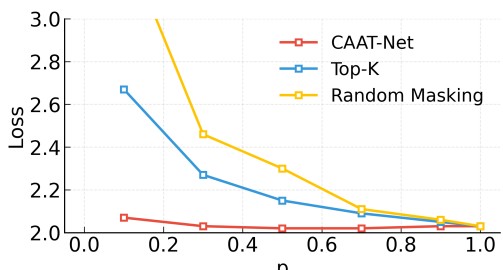

Figure 5: **Comparison to compression methods.** Validation loss vs. $p$ for 130M models using CAAT-Net, Top-k masking and random masking.

the same mask is applied in the backward pass as well. Furthermore, the Top-K and random masks are less efficient in reducing communication than CAAT-Net. See Appendix D for details.

To be consistent with CAAT-Net experiments, we denote by $p$ the probability that an activation is not zeroed (so if $p = 1$, applying the mask is equivalent to applying the identity function). We train the 130M parameter Llama model detailed in Section 5.1 for a total of 2.6B tokens. We experiment with different values of $p$, and report significant degradation for Top-K and random masks. Results are available in Figure 5.

## 6 Conclusion

In this paper, we show that with minor changes to current training frameworks, activations do not need to be fully synchronized in tensor-parallel communication for training to converge. We propose CAAT-Net, which significantly decreases tensor-parallel related network traffic in LLM training with minor to no degradation in accuracy. We experiment with smaller networks to explore the effects of tensor-parallel dimension and the synchronization factor. Finally, we show how our method speeds up inference and training workloads.

## Acknowledgments and Disclosure of Funding

The research of DS was funded by the European Union (ERC, A-B-C-Deep, 101039436). Views and opinions expressed are however those of the author only and do not necessarily reflect those of the European Union or the European Research Council Executive Agency (ERCEA). Neither the European Union nor the granting authority can be held responsible for them. DS also acknowledges the support of the Schmidt Career Advancement Chair in AI.

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

# A    Mathematical Description of MLP and Attention Using Partial Channel–Reduce

## A.1    Attention

In this section, we describe an attention layer using partial channel–reduce. Like in section 3.1, we examine a case where the tensor-parallel dimension is 2. Furthermore, for simplicity, we assume the model consists of only 2 attention heads.

**Standard all-reduce in Attention.** Below, we describe the output of the attention layer with a full all-reduce. The attention head outputs, $H_1$ and $H_2$, for the first and second heads, respectively, are

$$H_1 = \text{Attention}\left(XW_1^K, XW_1^Q, XW_1^V\right) \quad ; \quad H_2 = \text{Attention}\left(XW_2^K, XW_2^Q, XW_2^V\right). \tag{16}$$

In tensor-parallelism, the output projection of the attention layer $O$, is sharded between devices. The output $Z$ after all-reduce is:

$$Y = [H_1, H_2] \quad ; \quad Z = YO, \quad ; \quad O = \begin{bmatrix} O_1 \\ O_2 \end{bmatrix} \quad ; \quad Z = H_1O_1 + H_2O_2, \tag{17}$$

where $H_1O_1$ is calculated on the first device and $H_2O_2$ is calculated on the second one.

**Partial channel–reduce in Attention.** As in the MLP case, when introducing partial channel–reduce, the input to the attention is not necessarily identical on each device. The inputs on each device are denoted $X_1$ and $X_2$:

$$H_1 = \text{Attention}\left(X_1W_1^K, X_1W_1^Q, X_1W_1^V\right)$$
$$H_2 = \text{Attention}\left(X_2W_2^K, X_2W_2^Q, X_2W_2^V\right) \tag{18}$$

The outputs of the attention layer with partial channel–reduce are different per device, and are denoted $Z_1$ and $Z_2$:

$$Y = [H_1, H_2]. \quad ; \quad O = \begin{bmatrix} O_{11} & O_{21} \\ O_{12} & O_{22} \end{bmatrix} \tag{19}$$
$$Z_1 = [H_1O_{11} + H_2O_{12}, \ H_1O_{21}] \quad ; \quad Z_2 = [H_1O_{11} + H_2O_{12}, \ H_2O_{22}].$$

## A.2    Partial channel–reduce with many devices

In this section, we relax the assumptions made in Section 3.1 and Appendix A.1, to allow more than 2 tensor-parallel devices.

**MLP**. An MLP operation in transformers with a full all-reduce operation can be described as follows. Given weight matrices $A \in \mathbb{R}^{h \times f}$ and $B \in \mathbb{R}^{f \times h}$, where $f$ is the Feed Forward Network (FFN) hidden dimension, and input activations $X \in \mathbb{R}^{b \times t \times h}$, the MLP can be written in index notation as:

$$Z_{il} = \sum_k \sigma\left(\sum_j X_{ij}A_{jk}\right)B_{kl}, \tag{20}$$

where $t$ is the sequence length, $b$ is the batch size, and $\sigma$ is the nonlinearity. To introduce the partial channel–reduce operation, we first rewrite Eq. 20 in a notation that incorporates tensor-parallelism.

$$\tilde{Z}_{ilm} = \sum_k \sigma\left(\sum_j X_{ij}\tilde{A}_{jkm}\right)\tilde{B}_{klm}$$
$$Z_{il} = \sum_m \tilde{Z}_{ilm}, \tag{21}$$

where $\tilde{A}$ is $A$ written in column-parallel notation such that the last dimension corresponds to tensor-parallel rank, and similarly $\tilde{B}$ is $B$ written in row-parallel notation. The summation over $m$ is the all-reduce operation. Finally, we write the MLP with the partial channel–reduce operation as follows:

$$\tilde{Z}_{ilm} = \sum_k \sigma \left( \sum_j X_{ijm} \, \tilde{A}_{jkm} \right) \tilde{B}_{klm}$$

$$Z_{ilm} = \begin{cases} \sum_m \tilde{Z}_{ilm} & \text{if } l < \text{floor}\,(hp) \\ \tilde{Z}_{ilm} & \text{if } l \geq \text{floor}\,(hp) \end{cases} .$$

(22)

Here, the summation over $m$ is the partial channel–reduce operation. The main differences between Eq.21 and Eq.22 are:

- The sum over $\tilde{Z}$ is only over the first $p \cdot h$ channels of the hidden dimension.
- The input to the MLP is different per device when using partial channel–reduce, because the previous attention layer also used partial channel–reduce in its output. Thus, the input to MLP, $X$, has an additional index, $m$, corresponding to tensor-parallel device rank. Similarly, $Z$ has an additional index, $m$, because each device contains different parameters in the private channels after the partial channel–reduce operation.

**Attention**. We present the attention operation with partial channel–reduce. For simplicity, we assume that the tensor-parallel degree is the number of heads, though this assumption can easily be relaxed. The attention operation with a full operation is all-reduce is:

$$\mathrm{H}_{ijm} = \text{Attention} \left( XW_m^K, XW_m^Q, XW_m^V \right)_{ij}$$

$$\tilde{Z}_{ilm} = \sum_j \mathrm{H}_{ijm} \tilde{O}_{jlm}$$

$$Z_{il} = \sum_m \tilde{Z}_{ilm} \,,$$

(23)

where $H_{ijm}$ is an item in the $m^{th}$ attention head output, and $\tilde{O}_{jlm}$ is an item in the attention output projection, after reshaping the attention output projection $O$ to be written in a row-parallel notation. Hence, the output of the attention operation with the partial channel–reduce operation is:

$$\mathrm{H}_{ijm} = \text{Attention} \left( X_m W_m^K, X_m W_m^Q, X_m W_m^V \right)_{ij}$$

$$\tilde{Z}_{ilm} = \sum_j \mathrm{H}_{ijm} O_{jlm}$$

$$Z_{ilm} = \begin{cases} \sum_m \tilde{Z}_{ilm} & \text{if } l < \text{floor}\,(hp) \\ \tilde{Z}_{ilm} & \text{if } l \geq \text{floor}\,(hp) \end{cases} .$$

(24)

The main differences between Eq.23 and Eq.24 are:

- The sum over $\tilde{Z}$ is only over the first $p \cdot h$ channels of the hidden dimension.
- Apart from the first attention layer, the input to the attention is different per device when using partial channel–reduce, because the previous MLP layer also used partial channel–reduce in its output. Thus, the input to the attention, $X$, has an additional index, $m$, corresponding to the tensor-parallel rank. Similarly, $Z$ has an additional index because each device contains different parameters after the partial channel–reduce operation.

## B  Analyzing Forward-Backward Mismatch Considering Residual Connections

For a transformer with all-reduce, we examine the input to an MLP layer on device $m$, denoted $X_m$, as a function of the output of a previous attention layer on device $m$ before the all-reduce operation,

denoted $Z_m$ with $R$ being the residual connection from before the attention block, we have

$$X_m = \text{norm}\left(\sum_m Z_m + R\right).$$ (25)

$R$ is the residual connection from before the attention block. We look at the loss as a function of $X_m$ for all $m$, and of the residual connection (that skips the next layer) on the $m^{th}$ device, $R_m$. We also look at $X_m$ and $R_m$ as a function of $Z_1$, and calculate the derivative of the loss function w.r.t $Z_1$ using the chain rule:

$$\frac{\partial \mathcal{L}\left(X_1\left(Z_1\right),..,X_M\left(Z_1\right),R_1\left(Z_1\right),..,R_M\left(Z_1\right)\right)}{\partial Z_1} =$$
$$\sum_m \left(\frac{\partial \mathcal{L}\left(X_m\right)}{\partial X_m} \cdot \frac{\partial X_m\left(Z_1\right)}{\partial Z_1} + \frac{\partial \mathcal{L}\left(R_m\right)}{\partial R_m} \cdot \frac{\partial R_m\left(Z_1\right)}{\partial Z_1}\right) =$$
$$\sum_m \left(\frac{\partial \mathcal{L}\left(X_m\right)}{\partial X_m} \cdot \frac{\partial X_m\left(Z_1\right)}{\partial Z_1} + \frac{\partial \mathcal{L}\left(R_m\right)}{\partial R_m}\right) =$$
$$\sum_m \left(\frac{\partial \mathcal{L}\left(X_m\right)}{\partial X_m} \cdot \frac{\partial X}{\partial Z_1} + \frac{\partial \mathcal{L}\left(R_m\right)}{\partial R_m}\right)$$ (26)

where in the second equation we used the fact that

$$\frac{\partial R_m\left(Z_1\right)}{\partial Z_1} = \frac{\partial}{\partial Z_1}\sum_m Z_m = 1\,,$$ (27)

and in the last equation, we equated $\frac{\partial X_m}{\partial Z_1}$ for all $m$ (denoted now as $\frac{\partial X}{\partial Z_1}$) because in Tensor-Parallelism with full all-reduce operations $X_1\left(Z_1\right) = X_2\left(Z_1\right) = ... = X_M\left(Z_1\right)$, and thus their derivative w.r.t $Z_1$ are also equal. Here, the summation over $m$ is the all-reduce operation in the backward pass. Our key insight is here is that, due to the distributivity of matrix multiplication:

$$\sum_m \left(\frac{\partial \mathcal{L}\left(X_m\right)}{\partial X_m} \cdot \frac{\partial X}{\partial Z_1} + \frac{\partial \mathcal{L}\left(R_m\right)}{\partial R_m}\right) = \sum_m \left(\frac{\partial \mathcal{L}\left(X_m\right)}{\partial X_m}\right) \cdot \frac{\partial X}{\partial Z_1} + \sum_m \frac{\partial \mathcal{L}\left(R_m\right)}{\partial R_m}\,.$$ (28)

In other words, the all-reduce operation can happen before back-propagating through the normalization function (right side of the equation), or after the normalization function (left side of the equation). On the right hand side of the equation, the summation of the residual gradient happens in the all-reduce of the next layer, i.e the layer after the residual is accumulated back into the activation tensors. In standard training frameworks, such as Megatron-LM, the all-reduce operation happens before back-propagating through the normalization function. When training with CAAT-Net, the assumption that $\frac{\partial X_m}{\partial Z_m}$ are equal for all $m$ no longer holds. For this reason, the all-reduce operation in the backward pass must be after the calculation of the normalization function derivative.

Similarly, we examine the normalization function parameter update, denoted $\beta$, with a full all-reduce operation. Because the residual splits from the activations before the normalization function is applied, the normalization function parameter update is not a function of the residual states:

$$\frac{\partial \mathcal{L}\left(X_1\left(\beta\right),...,X_M\left(\beta\right)\right)}{\partial \beta} = \sum_m \left(\frac{\partial \mathcal{L}\left(X_m\right)}{\partial X_m} \cdot \frac{\partial X_m}{\partial \beta}\right)\,.$$ (29)

In the gradient descent step, this means that after each iteration, before updating the weights, the updates $\partial \beta$ need to be synchronized themselves with an all-reduce operation. In popular training frameworks, the fact that $\frac{\partial X_m}{\partial \beta}$ are equal for all $m$ is utilized once again. When the all-reduce happens before backpropagating through the normalization function, there is no need for gradient synchronization after each step:

$$\sum_m \left(\frac{\partial \mathcal{L}\left(X_m\right)}{\partial X_m}\right) \cdot \frac{\partial X}{\partial \beta}\,.$$ (30)

Once again, equated $\frac{\partial X_m}{\partial \beta}$ for all $m$ (denoted now as $\frac{\partial X}{\partial \beta}$).

## C   Speedup Analysis

We suggest a simple model to further understand the relation between compute and communication times, and analyze the potential speed-up of our method. We make the following assumptions:

- Compute is modeled only as GEMM operations — specifically in the MLP and attention. Communication is modeled only as the tensor-parallel all-reduce. We neglect all other computations (embeddings, language model head, etc.) and communications (pipeline parallel, data parallel, etc.)

- We assume a constant ratio between compute and communication capacity, $C$, given in units of $\frac{FLOP/s}{GB/s}$.

- We assume that the communication time roughly does not change when we adjust the tensor-parallel dimension. This assumption is true in Gaudi processors when using multi-node connections, due to inter- and intra-node communications overlapping during all-reduce operations. Additionally, we assume that computation time is inversely proportional to the tensor-parallel dimension. This assumption is only approximately true since computation time depends on many factors, such as tensor shapes.

We examine the forward pass of a transformer with a simple MLP, where the first weight matrix in the MLP is of dimension $h \times 4h$ and the second weight matrix is of dimension $4h \times h$. This type of MLP is common in LLM's such as GPT and BERT. We consider batch size 1, as batch size does not affect the results in this section. When multiplying an $m \times p$ matrix by a $p \times n$ matrix, the total number of operations (multiplications and additions) is $m \cdot p \cdot (2n - 1)$, which we approximate to $2m \cdot p \cdot n$. Based on this, the total number of computation operations in a transformer layer with sequence length $s$, $n$ heads and hidden size $h$ is:

$$\text{MLP ops} = \underbrace{8sh^2}_{\text{h to 4h}} + \underbrace{8sh^2}_{\text{4h to h}} = 16sh^2$$

$$\text{Attention ops} = \underbrace{6sh^2}_{\text{qkv proj}} + \underbrace{4s^2h}_{\text{attention}} + \underbrace{2sh^2}_{\text{out proj}} = \qquad (31)$$

$$4s^2h + 8sh^2 \,.$$

To calculate the total communication payload, we recall that in each full all-reduce operation, $s \cdot h$ activations are sent and received by each device, and this happens twice every transformer block. Thus the total number of compute operations $G$, and communication payload $P$ as a function of $p$, per device, are:

$$G = \frac{24sh^2 + 4s^2h}{r}$$

$$\qquad (32)$$

$$P(p) = 2shp \,,$$

where $r$ is the tensor-parallel dimension and $p$ is the CAAT-Net synchronization factor. Consequently, the total time for computation and communication of a single transformer layer in the forward pass is

$$T(p) = G + P(p) \,. \qquad (33)$$

We find that the speed-up introduced by our method is:

$$\frac{T(1) - T(p)}{T(1)} = (1 - p)\frac{1}{1 + \frac{12h+2s}{Cr}} \,. \qquad (34)$$

Our method is more significant for large $C$ and $r$, but diminishes for very large $h$ and $s$. This is because computation is quadratic in $h$ and $s$ while the communication is linear. If communication is hidden behind all of the computation, then there is a maximal value of $p$ at which communication and computation times are equal, and there is less motivation to decrease $p$ to improve speed-up. If $G \geq P(1)$, then the optimal value of $p$ is $p^* = 1$. Otherwise we can set $G = P(p)$. Considering both cases, we find that the value of $p^*$ is:

$$p^* = \min(\frac{12h + 2s}{Cr}, 1) \,. \qquad (35)$$

# D    Top-K and Random Masks Communication Reduction

Analyzing the communication volume reduction in these experiments, we note that only the communication in the forward pass is compressed. This is because we use vanilla tensor-parallelism, and not our alternate implementation detailed in Section 4, so communication doesn't happen in the same location in the forward and backward passes. For this reason, unlike CAAT-Net, communication compression in the forward pass does not imply communication compression in the backward pass. Additionally, all-reduce happens in 2 steps – reduce-scatter and all-gather. Applying a mask before communication is equivalent to compressing the reduce-scatter operation, which accounts for only half of the total communication in the all-reduce. For these reasons, compressing communication using a Top-K or random masking with parameter $p$ leads to a $(\frac{100 \cdot (1-p)}{4})\%$ reduction in tensor-parallel communication volume. In CAAT-Net with parameter $p$ communication is compressed in the reduce-scatter and all-gather of both the forward and backward passes. The total tensor-parallel communication in CAAT-Net is reduced by $(100 \cdot (1-p))\%$.

# E    Additional Results

## E.1    Effects of All-Reduce Bit Precision in the Backward Pass

In this section we show that 16 bit accumulation in the backward pass all-reduce leads to loss divergence. To show this, we train the variants of the TinyLlama model, as is described in Section 5.1. The first is trained using the Intel's Megatron-LM fork repository, without any additional changes. The second variant is trained after applying the changes detailed in Section 4, with the backward pass all-reduce accumulation in bfloat16 precision. The third is trained after applying all changes in the Section 4, including 32 bit accumulation in the backward pass. Results in Figure 6 show severe degradation with 16 bit accumulation after 15K training steps. Despite the mathematical equivalence between the two implementations, numerical differences lead to instability.

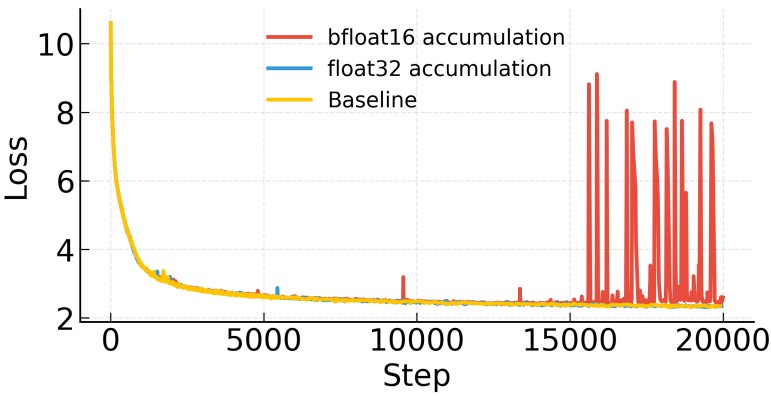

Figure 6: **Effects of All-Reduce Accumulation Precision in the Backward Pass on TinyLlama Train Loss**.

## E.2    TinyLlama and GPT-XL Accuracy Evaluation for varying values of $p$

We report the full zero-shot accuracy evaluation of TinyLlama using varying values of $p$. The evaluation metrics are those described in Section 5.1. Overall $p = 0.75$ achieves the best accuracy, with the highest score in 5 out of 8 metrics. Results are available in Table 1. Additionally, we report full zero-shot accuracy evaluation of GPT-XL for $p = 0.5$. Results are available in Table 2.

Table 2: **Zero-shot accuracy results on GPT-XL**

| $p$ | LAMBADA (acc) | Hellaswag (acc) | WinoGrande (acc) | PIQA (acc) |
|---|---|---|---|---|
| 1 | **50.16 ± 0.70** | 36.07 ± 0.48 | 52.41 ± 1.40 | **68.06 ± 1.11** |
| 0.5 | 48.17 ± 0.69 | **36.26 ± 0.48** | **53.83 ± 1.40** | 67.79 ± 1.09 |
| | OpenBookQA (acc) | BOOL-Q (acc) | WikiText (ppl) | Validation Loss |
| 1 | **19.20 ± 1.76** | 60.85 ± 0.85 | 19.25 | 1.28 |
| 0.5 | 18.60 ± 1.74 | **61.01 ± 0.85** | **19.00** | **1.27** |

Table 3: **Zero-shot accuracy results on TinyLlama for varying values of $p$**

| $p$ | LAMBADA (acc) | Hellaswag (acc) | WinoGrande (acc) | PIQA (acc) |
|---|---|---|---|---|
| 1 | 48.26 ± 0.70 | 35.55 ± 0.48 | **55.25 ± 1.40** | 67.03 ± 1.10 |
| 0.875 | 46.43 ± 0.69 | 35.89 ± 0.48 | 53.04 ± 1.40 | 64.96 ± 1.11 |
| 0.75 | **49.39 ± 0.70** | 35.88 ± 0.48 | 52.80 ± 1.40 | **68.99 ± 1.08** |
| 0.625 | 48.92 ± 0.70 | 35.61 ± 0.48 | 53.20 ± 1.40 | 67.41 ± 1.09 |
| 0.5 | 48.01 ± 0.70 | **35.65 ± 0.48** | 51.26 ± 1.40 | 67.68 ± 1.09 |
| 0.375 | 45.74 ± 0.69 | 35.48 ± 0.48 | 52.88 ± 1.40 | 68.28 ± 1.09 |
| 0.25 | 48.83 ± 0.70 | 35.63 ± 0.48 | 50.83 ± 1.40 | 67.74 ± 1.09 |
| 0.125 | 44.25 ± 0.69 | 34.63 ± 0.47 | 50.83 ± 1.38 | 66.27 ± 1.10 |
| 0 | 33.96 ± 0.66 | 30.49 ± 0.46 | 50.28 ± 1.41 | 64.09 ± 1.12 |
| | OpenBookQA (acc) | BOOL-Q (acc) | WikiText (ppl) | Validation Loss |
| 1 | 21.20 ± 1.83 | 59.57 ± 0.86 | 19.10 | 1.28 |
| 0.875 | 20.00 ± 1.79 | 60.24 ± 0.86 | 19.05 | 1.28 |
| 0.75 | 20.60 ± 1.81 | **61.04 ± 0.85** | **18.84** | **1.27** |
| 0.625 | **21.40 ± 1.84** | 60.12 ± 0.86 | 19.08 | 1.28 |
| 0.5 | 21.00 ± 1.82 | 59.24 ± 0.86 | 20.12 | 1.28 |
| 0.375 | 19.40 ± 1.77 | 57.13 ± 0.85 | 19.30 | 1.29 |
| 0.25 | 20.60 ± 1.81 | 51.53 ± 0.87 | 19.60 | 1.29 |
| 0.125 | 20.60 ± 1.81 | 54.07 ± 0.87 | 20.36 | 1.31 |
| 0 | 17.80 ± 1.71 | 56.97 ± 0.87 | 28.37 | 1.49 |

### E.3 Private Channel Scaling Ablations

In this section we present an ablation study for private channel scaling. We train the 130M model presented in Section 5.1 over 7.8B tokens, with a tensor-parallel dimension of 8 and $p = 0.5$. We record the validation loss at the end of training, with and without private channel scaling, and see a slight improvement when using private-channel scaling over most values of $p$. Results are presented in Figure E.3.

### E.4 Additional Speedup Results

In this section, we report inference speedup on 34B and 70B Llama models. These models use grouped query attention (GQA) of 8, so it is possible to evaluate our speedup only on tensor-parallel dimension of 8 (our a lower multiple of 2). Absolute measurements are shown in Tables 4 and 5.

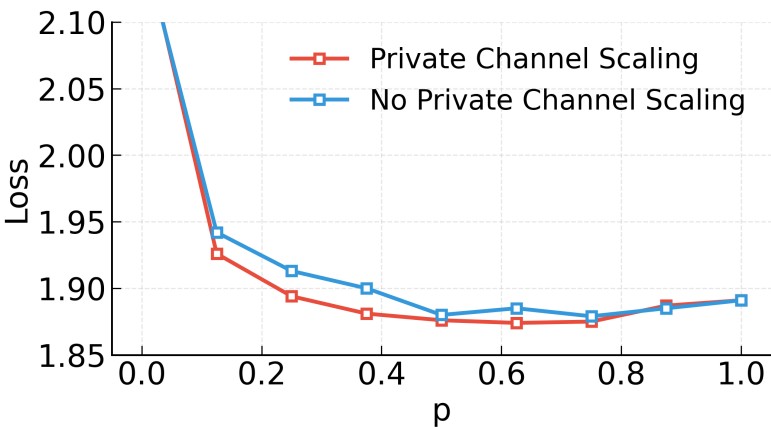

Figure 7: **Partial Channel Scaling**. Validation loss vs. $p$ for with and without private channel scaling, on a 130M model.

Table 4: **Absolute measurements of Llama 70B inference Time-to-First-Token (TTFT) for tensor-parallel 8.** Measured in milliseconds, with varying batch sizes and values of $p$.

| Batch size | Baseline | $p = 0.75$ | $p = 0.5$ | $p = 0.25$ | No communication |
|---|---|---|---|---|---|
| 16 | 1402 | 1301 | 1224 | 1153 | 1093 |
| 32 | 2769 | 2595 | 2436 | 2281 | 2092 |
| 64 | 5513 | 5290 | 4982 | 4654 | 4259 |
| 128 | 10962 | 10293 | 9817 | 9457 | 8731 |
| 256 | 20606 | 19242 | 18055 | 16854 | 15107 |

Table 5: **Absolute measurements of Llama 34B inference Time-to-First-Token (TTFT) for tensor-parallel 8.** Measured in milliseconds, with varying batch sizes and values of $p$.

| Batch size | Baseline | $p = 0.75$ | $p = 0.5$ | $p = 0.25$ | No communication |
|---|---|---|---|---|---|
| 16 | 730 | 682 | 637 | 590 | 532 |
| 32 | 1519 | 1429 | 1328 | 1226 | 1074 |
| 64 | 2977 | 2861 | 2675 | 2453 | 2178 |
| 128 | 6081 | 5738 | 5424 | 5081 | 4544 |
| 256 | 11104 | 10270 | 9575 | 8870 | 7942 |

### E.5 Absolute Performance Metrics

In Section 5.2 of the paper we reported speedup in training and inference on a Llama-7b model, relative to the baseline. Here we report the absolute measurements, from which the speedup was calculated.

Table 6: **Absolute measurements of training throughput.** Measured in tokens per second (tps) for different tensor-parallel (TP) dimensions, and values of $p$.

| TP dimension | Baseline | $p = 0.75$ | $p = 0.5$ | $p = 0.25$ | No communication |
|---|---|---|---|---|---|
| 4 | 34904 | 36883 | 38586 | 39728 | 40944 |
| 8 | 64395 | 67495 | 70403 | 72917 | 76646 |
| 16 | 96642 | 98625 | 106050 | 112456 | 126227 |

Table 7: **Absolute measurements of inference Time-to-First-Token (TTFT) for tensor-parallel 16.** Measured in milliseconds, with varying batch sizes and values of $p$.

| Batch size | Baseline | $p = 0.75$ | $p = 0.5$ | $p = 0.25$ | No communication |
|---|---|---|---|---|---|
| 16 | 214 | 209 | 187 | 165 | 100 |
| 32 | 443 | 386 | 358 | 326 | 193 |
| 64 | 746 | 696 | 637 | 565 | 391 |
| 128 | 1388 | 1291 | 1174 | 1060 | 792 |
| 256 | 2625 | 2393 | 2178 | 2059 | 1630 |

Table 8: **Absolute measurements of inference Time-to-First-Token for tensor-parallel 8.** Measured in milliseconds, with varying batch sizes and values of $p$.

| Batch size | Baseline | $p = 0.75$ | $p = 0.5$ | $p = 0.25$ | No communication |
|---|---|---|---|---|---|
| 16 | 204 | 189 | 181 | 160 | 139 |
| 32 | 411 | 374 | 352 | 320 | 279 |
| 64 | 853 | 786 | 727 | 641 | 571 |
| 128 | 1645 | 1571 | 1475 | 1295 | 1133 |
| 256 | 3320 | 3071 | 2916 | 2713 | 2432 |

### E.6 Measurements on NVIDIA Hardware

In this section we present the full inference speedup measurements on H100 and A100, as detailed in Section 5.2 of the paper.

Table 9: **TTFT on H100 and A100**. Measured in milliseconds, for different batch sizes and $p$ values.

| Device | Batch Size | $p = 1.0$ | $p = 0.75$ | $p = 0.5$ | $p = 0.25$ | $p = 0.0$ |
|---|---|---|---|---|---|---|
| **H100** | 16 | 173 | 166 | 157 | 149 | 136 |
| | 32 | 341 | 338 | 312 | 296 | 270 |
| | 64 | 679 | 663 | 623 | 589 | 545 |
| | 128 | 1360 | 1295 | 1280 | 1179 | 1088 |
| **A100** | 16 | 406 | 415 | 373 | 353 | 328 |
| | 32 | 795 | 764 | 727 | 694 | 643 |
| | 64 | 1568 | 1508 | 1436 | 1363 | 1281 |
| | 128 | 3117 | 2999 | 2927 | 2722 | 2553 |

