# OpenReview forum: "Tensor-Parallelism with Partially Synchronized Activations"
_NeurIPS.cc/2025/Conference — NeurIPS 2025 poster_

### Official Review · Reviewer_7RQG · 2025-06-23

**Clarity:** 3
**Significance:** 3
**Originality:** 3
**Rating:** 4
**Confidence:** 5

**Summary:**

This paper introduces CAAT-Net, a method for LLM training and inference ini a distributed setup (tensor parallelism) in a more efficient way that reduces communication overhead. Tensor-parallelism in LLMs requires extensive communication to synchronize activations across devices, which becomes a bottleneck as models scale. CAAT-Net addresses this by implementing partial channel-reduce where only a subset of activation channels are synchronized thereby decreasing bandwidth demands without less impacting model accuracy. To enable this optimization the paper proposes the architectural changes on synchronization point, including specific modifications to MLP and attention layers and gives implementation considerations like forward-backward mismatch and the necessity of fp32 accumulation for numerical stability. Authors state that CAAT-Net can achieve speedups in both training and inference for 1B and 7B parameter models.

**Questions:**

1. What if the normalization method applied on the transformer block is not RMSNorm? Is CAAT-net still applicable to that cases?

2. I’m confused about the statement that even though the implementations are mathematically identical, there is a large gap in training convergence. Why does this problem come up with 16bit precision although the two implementations are identical? If this problem is from the range of fp16, can bf16 also solve this problem?

**Ethical Concerns:**

["NO or VERY MINOR ethics concerns only"]

**Final Justification:**

The authors clearly resolved technical concerns regarding numeric precision, quality-performance relationship.
However, I still have a concern on evaluations of this paper (Lack of results: showing the results with validation loss, presenting only on zero-shot tasks).
Therefore, I raised score from 3 to 4 (still borderline).

**Limitations:**

yes

**Paper Formatting Concerns:**

The paper has no issues on formatting.

**Quality:**

2

**Strengths And Weaknesses:**

Strengths


1. While tensor parallelism provides more computability and larger scalability to model training, the distributed setup is often bottlenecked by communication between devices, e.g., all-reduce. This problem becomes significant in larger scale training working with a number of devices since the number of communication linearly increases. This paper addresses an important issue in the efficiency of training large models.

2. Clear background explanation about tensor parallelism with and without all-reduce operation


Weaknesses


Although the paper tries to address the important problem, I have a lot of concerns on evaluations.
1. The paper only shows accuracy results on p=0.5 setup on tensor parallel 8. While this setup well preserves the results compared to the baseline, it is not sufficient to say that all the speedups in Figure3 are meaningful results.
2. The evaluation on zero-shot tasks typically generate a single token which aren’t sufficient to validate the quality of trained model. Showing the impact on generation accuracy when the model is required to produce a longer sequence could firmly validate the effect of partial channel-reduce.
3. The evaluated models are limited to LLaMA2 and TinyLlama models. Testing on other recent models (e.g., Llama 3.3, Qwen 2.5, …) could validate its generalizability.

---

> ### Author Rebuttal · Authors · 2025-07-30
>
> We thank the reviewer for their careful evaluation and constructive comments. We appreciate that the reviewer recognizes the importance of the problem we aim to solve, and the significance of our approach. To address the reviewers concerns regarding evaluation, we expanded the setups in which we evaluate our model, including experimentation with additional architectures and  varying values of $p$.
>
> ---
>
> **(1)**
>
> **Question** - What if the normalization method applied on the transformer block is not RMSNorm? Is CAAT-net still applicable to that cases?
>
> **Answer** CAAT-Net is applicable to any normalization function, as the derivations in Section 4 are independent of the specific normalization method used. We used RMSNorm as a standard example, but the approach generalizes to any normalization function. The calculations in Section 4 should indeed be presented with a general normalization function for clarity. We will update Section 4 to use general notation in the camera-ready version. Futhermore, we have added training experiments with GPT3-XL [A] using LayerNorm to demonstrate this generalizability (see section 5 of this rebuttal).
>
> ---
>
> **(2)**
>
> **Question** - I'm confused about the statement that even though the implementations are mathematically identical, there is a large gap in training convergence. Why does this problem come up with 16bit precision although the two implementations are identical? If this problem is from the range of fp16, can bf16 also solve this problem?
>
> **Answer** Although the two implementations are mathematically equivalent, they are not identical when implemented in finite precision. The changed order of floating-point operations introduces different rounding errors that compound during training. We find that when moving the all-reduce in the backward pass, the errors that accumulate throughout training are significant, and using fp32 accumulation solves this issue.
>
> Regarding bf16: our baseline experiments use bf16, and we observed the convergence issues in this setting.
>
> ---
>
> **(3)**
>
>  **Weakness** - The paper only shows accuracy results on p=0.5 setup on tensor parallel 8. While this setup well preserves the results compared to the baseline, it is not sufficient to say that all the speedups in Figure 3 are meaningful results.
>
> **Response** - In figure 3 we ran multiple experiments which don't require full model training. Running all of the experiments in figure 3 on large models would require an unrealistic amount of time and computation resources. However, to address this concern, we have run a few additional experiments. We hope that this convinces the reviewer that CAAT-Net achieves good accuracy in multiple settings, which we see robustly in our experimentation. First, we replaced Table 2 in our paper with a graph to include a more detailed visualization of the trends in $p$ and TP for a 130M model. Unfortunately, we cannot share the graph in this rebuttal due to the guidelines, so it is presented here as a table. (The results are slightly different than those in the paper because a different learning rate scheduler was used):
>
> | $p$    | $1$    | $0.9$  | $0.8$  | $0.7$  | $0.6$  | $0.5$  | $0.4$  | $0.3$  | $0.2$  | $0.15$ | $0.1$  | $0.05$ | $0.03$ | $0.01$ | $0.001$ |
> |--------|--------|--------|--------|--------|--------|--------|--------|--------|--------|--------|--------|--------|--------|--------|---------|
> | TP=2   | 1.887  | 1.890  | 1.890  | 1.883  | 1.879  | 1.876  | 1.872  | **1.870**  | 1.871  | 1.874  | 1.878  | 1.880  | 1.889  | 1.899  | 1.970   |
> | TP=4   | 1.884  | 1.884  | 1.882  | 1.879  | 1.873  | 1.867  | **1.863**  | 1.872  | 1.878  | 1.884  | 1.893  | 1.916  | 1.924  | 1.952  | 2.056   |
> | TP=8   | 1.889  | 1.892  | 1.879  | 1.875  | 1.871  | **1.876**  | 1.880  | 1.886  | 1.895  | 1.904  | 1.920  | 1.950  | 1.978  | 2.020  | 2.152   |
> | TP=16  | 1.887  | 1.880  | 1.879  | 1.877  | **1.872**  | 1.878  | 1.886  | 1.903  | 1.918  | 1.931  | 1.953  | 1.992  | 2.011  | 2.059  | 2.264   |
>
>
> The best validation loss per tensor-parallel configuration is highlighted. The reviewer can see training accuracy remains intact, and can even improve, for large enough values of $p$.
>
> Additionally, we conduct additional experiments with varying values of $p$ on TinyLlama to show that these trends scale relatively well. In these experiment TinyLlama was trained over 45B tokens. The best value per task is highlighted.
>
> | **Model** | **LAMBADA (acc)**     | **Hellaswag (acc)**   | **WinoGrande (acc)**   | **PIQA (acc)**        | **OpenBookQA (acc)**  | **BOOL-Q (acc)**      | **WikiText (ppl)** | **Validation Loss** |
> |-----------|------------------------|------------------------|-------------------------|------------------------|------------------------|------------------------|---------------------|----------------------|
> | $p=1$ | **43.59 ± 0.69**           | 33.41 ± 0.47           | 51.22 ± 1.40            | 66.65 ± 1.10           | 17.80 ± 1.71           | **59.05 ± 0.86**           | **21.44**              | **1.36**                 |
> | $p=0.7$ | 42.73 ± 0.69           | **33.59 ± 0.47**           | 51.70 ± 1.40            | 66.05 ± 1.10           | 18.20 ± 1.73           | 58.17 ± 0.86           | 21.50              | **1.36**                 |
> | $p=0.5$ | 42.64 ± 0.69           | 33.04 ± 0.47           | 51.70 ± 1.40            | **67.25 ± 1.09**           | 17.60 ± 1.70           | 56.57 ± 0.87           | 22.66              | 1.37                 |
> | $p=0.3$ | 40.33 ± 0.68           | 33.62 ± 0.47           | **52.80 ± 1.40**        | 66.76 ± 1.10           | **19.40 ± 1.77**       | 56.97 ± 0.87           | 21.80              | 1.37                 |
> | $p=0.1$ | 39.94 ± 0.68           | 33.14 ± 0.47           | 51.22 ± 1.40            | 66.32 ± 1.10           | 17.20 ± 1.69           | 56.97 ± 0.87           | 22.75          | 1.40             |
>
> ---
>
> **(4)**
>
> **Weakness** The evaluated models are limited to LLaMA2 and TinyLlama models. Testing on other recent models (e.g., Llama 3.3, Qwen 2.5, ...) could validate its generalizability.
>
> **Response** To address this concern, we trained a version of the 1.3B parameter GPT-XL with partial channel-reduce. As per the specific requests for Qwen and Llama3 - both have an architecture which is very similar to Llama2, so training GPT3-XL is a better candidate to show that our method is robust. We trained GPT3-XL with using tensor-parallel 8, on the RedPajama dataset using the GPTSentencePiece tokenizer, rotary positional embeddings and with a global batch size of 512. We trained for a total of 45B tokens. Similarly to results on Llama, we find that CAAT-Net maintains and even slightly improves accuracy results.
>
> | **Model** | **LAMBADA (acc)**     | **Hellaswag (acc)**   | **WinoGrande (acc)**   | **PIQA (acc)**        | **OpenBookQA (acc)**  | **BOOL-Q (acc)**      | **WikiText (ppl)** | **Validation Loss** |
> |-----------|------------------------|------------------------|-------------------------|------------------------|------------------------|------------------------|---------------------|----------------------|
> | $p=1$     | **50.15 ± 0.70**       | 35.93 ± 0.48           | 52.49 ± 1.40            | 67.41 ± 1.09           | **21.4 ± 1.84**        | **61.74 ± 0.85**       | 19.58              | 1.30                 |
> | $p=0.5$   | 49.56 ± 0.70           | **36.27 ± 0.48**       | **53.20 ± 1.40**        | **68.06 ± 1.09**       | 20.80 ± 1.82           | 59.66 ± 0.86           | **19.35**          | **1.29**             |
>
> ---
>
> **(5)**
>
> **Weakness** The evaluation on zero-shot tasks typically generate a single token which aren't sufficient to validate the quality
> of trained model. Showing the impact on generation accuracy when the model is required to produce a longer sequence could firmly validate the effect of partial channel-reduce.
>
> **Response** We thank the reviewer for the valuable suggestion to evaluate longer sequence generation. While we acknowledge that multi-token generation can offer additional insights, we focused on single-token predictions to align with the standard zero-shot evaluation protocols widely used in the literature. Established methods such as SparseGPT [B] , SmoothQuant [C], and recently Ladder Residual [12] (to name a few) have all been evaluated using solely this approach. In our view, single-token evaluations provide a reliable and well-established proxy for assessing model quality.
>
> ---
>
> [A] Tom Brown, Benjamin Mann, Nick Ryder, Melanie Subbiah, Jared D Kaplan, Prafulla Dhariwal, Arvind Neelakantan, Pranav Shyam, Girish Sastry, Amanda Askell, et al. Language models are few-shot learners. Advances in neural information processing systems, 33:1877–1901, 2020.
>
> [B] E. Frantar and D. Alistarh. 2023. Massive language models can be accurately pruned in one-shot. arXiv preprint
> arXiv:2301.00774.
>
> [C] Guangxuan Xiao, Ji Lin, Mickael Seznec, Hao Wu, Julien Demouth, and Song Han. 2023. Smoothquant: Accurate and efficient post-training quantization for large language models. In International Conference on Machine Learning, pages 38087–38099. PMLR.

---

> > ### Comment · Reviewer_7RQG · 2025-08-02
> >
> > Thanks for providing answers on the questions.
> >
> > (2) It would be better to state in Section4 that bf16 was used instead of 16bit precision since this is pretty important detail.
> >
> > (3) How about providing speed ups (in terms of TTFT and training time of 1 step) at least in this results? Since there can be many combinations for one model due to diverse level of 'TP' and 'p', the more meaningful result this paper can provide to readers should be "How much quality degradation should we endure to get some amount of performance benefit in some level of TP?" in addition to "Cutting the 50% amount of communication in 8 devices can give 10~15% performance gain without quality degradation."
> >
> > (5) Isn't this case different from SparseGPT, SmoothQuant, and Ladder Residual which are applying their methods on top of the pretrained model while CAAT-net is training from scratch?

---

> > > ### Author Response · Authors · 2025-08-05
> > >
> > > We thank the reviewer for the time taken to read our response and provide thoughtful feedback. Below, we discuss the remaining comments and concerns.
> > >
> > > ---
> > >
> > > **Question**
> > >
> > > (2)  It would be better to state in Section4 that bf16 was used instead of 16bit precision since this is pretty important detail.
> > >
> > > **Answer**
> > >
> > > We thank the reviewer for pointing this out. We will update the paper for the camera-ready version.
> > >
> > > ---
> > >
> > > **Question**
> > >
> > > (3) How about providing speed ups (in terms of TTFT and training time of 1 step) at least in this results? Since there can be many combinations for one model due to diverse level of 'TP' and 'p', the more meaningful result this paper can provide to readers should be "How much quality degradation should we endure to get some amount of performance benefit in some level of TP?" in addition to "Cutting the 50\% amount of communication in 8 devices can give 10~15\% performance gain without quality degradation."
> > >
> > > **Answer**
> > >
> > > To experiment robustly with TP and p's effects on accuracy, we must fully train a model multiple times. This is possible for relatively small models (130M, 1B). Speedup, however, should be measured on larger models (7B, 70B, etc.). It is not feasible to train these large models multiple times. For this reason, our accuracy and speedup analysis mostly don't focus on the same models. Find speedup for 34B and 70B models in Appendix D.1 (in the supplementary materials).
> > >
> > > In response to this specific request, the TTFT results are presented below. During this process, we observed issues arising when the number of reduced channels is not divisible by 16 (i.e. when $p$ is not a multiple of $2^{-7}$). This is a technical, hardware-dependent issue, and we will make additional efforts to solve it. Furthermore, even without solving this problem, it does not significantly limit our method as there are still many possible choices of $p$ to choose from.
> > >
> > > Unfortunately, the chosen values $p=0.1, 0.3, 0.7$ are not multiples of $2^{-7}$, so we did not get good speedup for these specific numbers. For that reason, we included speedup results for the closest multiples of $2^{-7}$ which are  $p=0.1015625, 0.296875, 0.703125$. Results are presented in the graph below in ms. We emphasized $p=0.1015625, 0.296875, 0.5, 0.703125, 1$, for which there is monotonic improvement in TTFT.
> > >
> > > | **Batch Size** | **p=0.1** | **p=0.1015625** | **p=0.296875** | **p=0.3** | **p=0.5** | p=0.7 | **p=0.703125** | **p=1.0** |
> > > |----------------|---------|---------|----------|---------|---------|---------|----------|---------|
> > > | 16             |  78.56   | **65.69**  | **69.32**  | 79.66   | **73.30**   | 89.04   | **81.46**    | **84.57**   |
> > > | 32             | 151.95   | **127.97**  |  **133.71**  | 158.09  | **140.63**  | 174.20  | **153.53**   | **160.12**  |
> > > | 64             | 282.89  | **248.65** |  **250.27** | 306.29  | **263.74**  | 339.14  | **283.36**   | **311.99**  |
> > > | 128            | 569.83  | **466.13**  |  **496.46**  | 601.31  | **526.79**  | 681.43  | **585.14**   | **610.74**  |
> > > | 256            | 1123.88  | **936.30** | **1009.66** | 1168.93 | **1017.78** | 1294.47 | **1116.78**  | **1161.66** |
> > >
> > >
> > > We understand the reviewers requests regarding trade-off between accuracy and speedup. We hope that this sufficiently answers the reviewers question. Furthermore, we will provide a full graph after fixing this issue, with additional values of $p$, for the camera-ready version.
> > >
> > > ---
> > >
> > > **Question**
> > >
> > > (5) Isn't this case different from SparseGPT, SmoothQuant, and Ladder Residual which are applying their methods on top of the pretrained model while CAAT-net is training from scratch?
> > >
> > > **Answer**
> > >
> > > The reviewer is correct about SparseGPT and SmoothQuant. However, to our understanding,  the reviewer's claims that: "Showing the impact on generation accuracy when the model is required to produce a longer sequence could firmly validate the effect of [your method]." should apply equally to pre-training and post-training techniques.
> > >
> > > Furthermore, in the Ladder Residual paper, they both experiment with training from scratch and with fine-tuning. When training from scratch, they evaluate with zero-shot tasks.
> > >
> > > Finally, there are additional examples of papers that evaluate using zero-shot tasks is common in other papers that pretrain models. See, for example [A], or Microsoft's work on one-bit LLM's [B], among many.
> > >
> > > ---
> > >
> > > [A] R. Wang, Y. Gong, X. Liu, G. Zhao, Z. Yang, B. Guo, Z. Zha,
> > > and P. Cheng, “Optimizing large language model training using fp4
> > > quantization,” arXiv preprint arXiv:2501.17116, 2025. 12, 13
> > >
> > > [B] Shuming Ma, Hongyu Wang, Lingxiao Ma, Lei Wang, Wenhui Wang, Shaohan Huang,
> > > Li Dong, Ruiping Wang, Jilong Xue, and Furu Wei. The era of 1-bit llms: All large
> > > language models are in 1.58 bits. arXiv preprint arXiv:2402.17764, 2024

---

> > > > ### Comment · Reviewer_7RQG · 2025-08-08
> > > >
> > > > Thanks a lot for providing answers on additional questions. The technical concerns I had on this paper are well resolved.
> > > >
> > > > Please, include our discussion in appendix session and consider the points below.
> > > >
> > > > * Explaining the limitation of choosing 'p' and the reason (requiring divisible number of reduced channel).
> > > >
> > > > * Including loss curve with bf16 accumulation if it exists.
> > > >
> > > > * Including performance on a generation task to enhance the effectiveness of CAAT

---

> > > > > ### Author Response · Authors · 2025-08-08
> > > > >
> > > > > Thank you for your helpful suggestions. We will incorporate them into the revision and hope these improvements are sufficient to increase the score.

---

### Official Review · Reviewer_8fqL · 2025-07-03

**Clarity:** 2
**Significance:** 3
**Originality:** 4
**Rating:** 4
**Confidence:** 3

**Summary:**

This paper introduces CAAT-Net, a novel approach to reduce the communication overhead in tensor-parallel training and inference of LLMs. Traditional tensor-parallelism requires full synchronization of activation tensors across devices using expensive all-reduce operations. CAAT-Net relaxes this constraint by synchronizing only a subset of activation channels. Thereby it reduces communication costs while maintaining model accuracy. The authors validate their method through extensive experiments on LLaMA2-7B, TinyLLaMA, and a smaller 130M-parameter model, and analyze how the synchronization factor and tensor-parallel degree affect performance.

**Questions:**

1. How well does CAAT-Net interact with other techniques such as quantization, gradient checkpointing, or activation compression?
2. You mention that inference must use the same tensor-parallel configuration and p as training. Is there a principled way to support variable p at inference time or to convert a CAAT-Net model back to a fully synchronized model (p = 1) without retraining?

**Ethical Concerns:**

["NO or VERY MINOR ethics concerns only"]

**Final Justification:**

I would give this paper a borderline accept because it is a complete work with easy-use design, sufficient experiments, and well-written manuscripts.

**Limitations:**

Please refer to weakness and questions.

**Paper Formatting Concerns:**

None.

**Quality:**

3

**Strengths And Weaknesses:**

Strengths:
1. This paper proposes an effective modification to tensor-parallelism that halves communication cost without requiring model compression or accuracy trade-offs.
2. Experiments on models of different scales and tasks showing comparable or improved zero-shot accuracy.
3. Benefits observed in both training and inference, which is crucial for deployment.

Weaknesses:
1. Requires non-trivial changes to standard tensor-parallel frameworks such as Megatron.
2. While results on 7B models are encouraging, it remains to be seen if the same gains hold for 100B+ parameter models, where communication bottlenecks are most severe.
3. Lack of experiments on Nvidia GPUs, which limited the generation of this paper.

---

> ### Author Rebuttal · Authors · 2025-07-30
>
> We thank the reviewer for taking the time to read our paper carefully and provide thoughtful feedback. We appreciate the recognition of the originality and significance of our work. Specifically, we value the acknowledgment of its effectiveness in reducing communication overhead without sacrificing accuracy, across both training and inference. Below, we address the reviewer’s concerns in detail. As requested, we have included experiments on Nvidia GPUs, which further strengthen our claims and demonstrate the broader applicability of our method. Furthermore, we refer the reviewer to large-scale speedup experiments that appear in the appendix.
>
> ----------
>
> **(1)**
>
> **Weakness**
> Requires non-trivial changes to standard tensor-parallel frameworks such as Megatron.
>
> **Response**
> Our method requires changing only a few lines of code (less than 50 for training, and less than 10 for inference). Additionally, our code will be open source upon acceptance. For these reasons, we do not view this to be a weakness.
>
> ----------
>
>  **(2)**
>
> **Weakness**
> While results on 7B models are encouraging, it remains to be seen if the same gains hold for 100B+ parameter models, where communication bottlenecks are most severe.
>
> **Response**
> If the concern is about speedup measurements on 100B+ models, we would like to refer the reviewer to Appendix D.1, where we report inference speedups for larger Llama models, including Llama-34B and Llama-70B. On Llama-70B, our method achieves a 13% speedup for $p=0.5$ and a 23% speedup for $p=0.25$, demonstrating that our approach remains effective even at larger scales. While we have not yet run evaluations on 100B+ models, we can run these experiments in the near future, before the camera-ready deadline.
>
> If the concern is about training-time accuracy on 100B+ models, we note that most prior works in the literature evaluate methods at smaller scales (e.g., 125M to 7B parameters), due to the prohibitive computational cost of training 100B+ models. For example, prominent works such as GaLore [A] and Rho1 [B] demonstrate their results primarily on models up to 7B parameters. Even training our 7B model requires substantial compute resources and a large cluster, which is beyond the reach of many academic groups. Training 100B+ models can cost hundreds of thousands of dollars.
>
> Nevertheless, we show in our main results that our method scales well from 130M to 7B parameters, consistently preserving accuracy while improving performance.
>
> ----------
>
> **(3)**
>
> **Weakness**
> Lack of experiments on Nvidia GPUs, which limited the generation of this paper.
>
> **Response**
> To address the reviewer’s concern and show the applicability of our method to other hardware, we conducted additional experiments with Nvidia GPUs. Our experiments were done using the gpt-fast [C] repository, and consisted of replacing all-reduce with partial channel-reduce during inference. We conducted experiments on both:
>
> -   8 × NVIDIA H100-80GB-HBM3 with NVLink
>
> -   8 × NVIDIA A100-SXM4-80GB with NVLink
>
>
> Experiments were conducted on Llama2-7B using tensor-parallel 8 and a sequence length of 2048.
>
> **H100: TTFT (s) and Relative Speedup vs. p=1.0:**
>
> | Batch Size | $p=1.0$           | $p=0.75$          | $p=0.5$           | $p=0.25$          | $p=0.0$           |
> | ---------- | ----------------- | ----------------- | ----------------- | ----------------- | ----------------- |
> | 16         | 0.173s, **1.00×** | 0.166s, **1.04×** | 0.157s, **1.09×** | 0.149s, **1.14×** | 0.136s, **1.22×** |
> | 32         | 0.341s, **1.00×** | 0.338s, **1.01×** | 0.312s, **1.09×** | 0.296s, **1.13×** | 0.270s, **1.21×** |
> | 64         | 0.679s, **1.00×** | 0.663s, **1.02×** | 0.623s, **1.08×** | 0.589s, **1.13×** | 0.545s, **1.20×** |
> | 128        | 1.360s, **1.00×** | 1.295s, **1.05×** | 1.280s, **1.06×** | 1.179s, **1.13×** | 1.088s, **1.20×** |
>
> **A100: TTFT (s) and Relative Speedup vs. p=1.0**
> | Batch Size | $p=1.0$           | $p=0.75$          | $p=0.5$           | $p=0.25$          | $p=0.0$           |
> | ---------- | ----------------- | ----------------- | ----------------- | ----------------- | ----------------- |
> | 16         | 0.406s, **1.00×** | 0.415s, **0.98×** | 0.373s, **1.08×** | 0.353s, **1.13×** | 0.328s, **1.19×** |
> | 32         | 0.795s, **1.00×** | 0.764s, **1.04×** | 0.727s, **1.09×** | 0.694s, **1.13×** | 0.643s, **1.19×** |
> | 64         | 1.568s, **1.00×** | 1.508s, **1.04×** | 1.436s, **1.08×** | 1.363s, **1.13×** | 1.281s, **1.18×** |
> | 128        | 3.117s, **1.00×** | 2.999s, **1.04×** | 2.927s, **1.06×** | 2.722s, **1.13×** | 2.553s, **1.18×** |
>
>
> We can see up to **9% speedup for $p=0.5$** and **14% speedup for $p=0.25$** on Nvidia hardware. Importantly, these results are preliminary and not fully optimized, and we believe they can be further improved.
>
> If the reviewer is asking about training accuracy, we point out that there should not be any significant discrepancy in accuracy between Nvidia and Intel hardware.
>
> ----------
>
> **(4)**
>
> **Question**
> How well does CAAT-Net interact with other techniques such as quantization, gradient checkpointing, or activation compression?
>
> **Answer**
> We appreciate the reviewer’s interest in how our method interacts with techniques such as quantization, gradient checkpointing, and activation compression. While these combinations are indeed important and promising directions for future work, it is not feasible within the scope of this study to comprehensively evaluate our method alongside the large and continually growing set of existing techniques. Our current work focuses on thoroughly establishing the core effectiveness of our method, and we leave exploration of its integration with other optimization or compression methods to future research.
>
> ----------
>
> **(5)**
>
> **Question**
> You mention that inference must use the same tensor-parallel configuration and p as training. Is there a principled way to support variable p at inference time or to convert a CAAT-Net model back to a fully synchronized model (p = 1) without retraining?
>
> **Answer**
> We appreciate the reviewer’s comment and the opportunity to clarify our wording.
>
> We state in the paper that changing the value of $p$ at inference time effectively alters the model architecture and requires fine-tuning. However, we do not claim that inference must strictly use the same tensor-parallel configuration as training. We only note that using the same configuration ensures the inference benefits from reduced communication compared to the $p=1$ baseline.
>
> If the tensor-parallel configuration differs at inference, there are two primary options:
>
> -   **Post-training adjustment of $p$** via fine-tuning, which allows transitioning back to $p=1$ (fully synchronized) or other values.
>
> -   **Logical TP mapping at inference**: An alternative is to simulate the training-time TP configuration during inference, without altering $p$. For instance, if the model was trained with TP=2 but needs to be served with TP=1, the single inference device can emulate multiple logical TP devices. In this case, partial channel-reduce operations at attention and MLP layers can be replaced with intra-device summation. While this approach avoids retraining, it may incur extra computation and memory usage on the device, and its efficiency depends on the specific deployment environment.
>
>
> In practice, we recommend maintaining the training-time tensor-parallel configuration for inference, though it is not strictly required. If the reviewer deems it necessary, we can update the paper to make this point clearer.
>
> ---
>
> [A] J. Zhao, Z. Zhang, B. Chen, Z. Wang, A. Anandkumar, and Y. Tian, “Galore: Memory-efficient llm training by gradient low-rank projection,” arXiv preprint arXiv:2403.03507, 2024.
>
> [B] Zhenghao Lin, Zhibin Gou, Yeyun Gong, Xiao Liu, Yelong Shen, Ruochen Xu, Chen Lin, Yujiu Yang, Jian Jiao, Nan Duan, et al. 2024b. Rho-1: Not all tokens are what you need. arXiv preprint arXiv:2404.07965.
>
> [C] PyTorch Labs. gpt-fast, 2024. URL https://github. com/pytorch-labs/gpt-fast. Accessed: 2024- 09-29.

---

> > ### Comment · Reviewer_8fqL · 2025-08-05
> > **Rebuttal reviewer response**
> >
> > The authors have provided satisfactory clarifications and demonstrations. I'll increase my score.

---

### Official Review · Reviewer_f4Cy · 2025-07-04

**Clarity:** 3
**Significance:** 2
**Originality:** 3
**Rating:** 4
**Confidence:** 5

**Summary:**

This paper proposes CAAT-Net, a Communication-Aware Architecture for Tensor-parallelism, designed to reduce the communication overhead inherent in training and inference of Large Language Models (LLMs). Traditional tensor-parallelism fully synchronizes activation tensors across devices, requiring substantial communication. CAAT-Net introduces partial channel–reduce operations that synchronize only a subset of channels, significantly reducing communication demands without compromising model accuracy. This design achieves up to a 50 % reduction in communication payload (as shown on Llama2-7B and TinyLlama) while preserving model accuracy. The paper’s channel-level communication analysis is innovative, but the manuscript would be strengthened by a more thorough empirical evaluation: additional metrics (e.g., end-to-end throughput, latency breakdowns, scalability across more model sizes) and ablation studies are needed to fully characterize CAAT-Net’s behavior under diverse workloads.

**Questions:**

1. **Overlap Efficiency Claim**
   - **Issue:** Partial synchronization is claimed to reduce computation–communication overlap.
   - **Request:** In my understanding, even the model is larger, we still have enough computation time to overlap with communication. Overlapping is still effective here. Please correct this part or show some more robust evidence here.

2. **Comparison to Desynchronized-Attention / Ladder Schemes**
   - **Issue:** No head-to-head comparison with existing ladder or desynchronized-attention approaches.
   - **Request:** Include a representative experiment (or cost model) contrasting bandwidth reduction vs. accuracy/throughput trade-offs between CAAT-Net and other methods (e.g., **Ladder-Residual**).

3. **Justification for Partial Synchronization**
   - **Issue:** It’s unclear why synchronizing only a subset of channels preserves correctness.
   - **Request:** Provide:
     - A theoretical argument (e.g., variance or information-flow analysis), or
     - An ablation study showing that removing random channel subsets up to fraction \(1 - p\) incurs < 0.1 % accuracy loss.
     - Author shall design more fine-grained algorithm.

4. **Channel-Selection Strategy & Figure 2.C**
   - **Issue:** Figure 2.C picks contiguous channels for all-reduce without justification.
   - **Request:** Add an ablation comparing contiguous vs. interleaved channel selection (e.g., a hybrid split into B\(_{11}\), B\(_{12}\), B\(_{13}\) with communication on B\(_{11}\) and B\(_{13}\)).

5. **Scalability & Inference Flexibility**
   - **Issues:**
     1. How does the optimal sync fraction \(p\) change for much larger models (30 B+ parameters)?
     2. Can inference run on a different device count than training without full fine-tuning?
   - **Requests:**
     - (a) Provide cost projections or small-scale experiments on a 30 B-parameter model.
     - (b) Propose a strategy (e.g., dynamic \(p\) adjustment or lightweight channel-wise calibration) to avoid complete retraining when device counts change.
     - (c) I don't think it's appropriate to put this part in future study.

**Ethical Concerns:**

["NO or VERY MINOR ethics concerns only"]

**Final Justification:**

The technical concerns I had with this paper are mostly resolved. I will raise my point.

But I hope to see the final manuscript include our discussion in the full paper/appendix and include the points below.

* Ladder-residual is also trained from scratch. I hope to see at least a discussion in the related work or a comprehensive analysis in the appendix.
* I am still doubting ‘we show no accuracy degradation’, please include more accuracy training results for larger models.

**Limitations:**

Partially yes. The authors acknowledge the limitations, particularly regarding inference flexibility. There is no so muchnegative societal impacts or broader generalizability issues would strengthen the paper. But The authors are encouraged to explicitly address whether partial synchronization could introduce biases or reduce robustness in specific use cases. And the authors shall compare related methods more directly to reveal this work's potential limitations.

**Paper Formatting Concerns:**

No formatting issues identified; the paper adheres well to NeurIPS formatting guidelines.

**Quality:**

3

**Strengths And Weaknesses:**

## Strengths

### Quality
- The paper rigorously formulates the **partial channel–reduce** operation for both MLP and attention layers.
- Numerical stability concerns (e.g., fp32 accumulation) are clearly identified and addressed in the implementation (§4).

### Clarity
- Visualizations highlight when and how the all-reduce shifts between forward and backward passes under partial synchronization.

### Significance
- By synchronizing only a fraction \(p\) of channels (e.g.\ \(p=0.5\)), CAAT-Net achieves up to **50 %** communication reduction and delivers **14 %** throughput gains on Gaudi3 clusters.

### Originality
- **Channel-level synchronization** is a novel axis.

---

## Weaknesses

### Quality
- **Figure redundancy**: Figures 1 and 2 convey very similar information. Merging or re-annotating them would improve flow.
- **Numerical details**: The impact of fp32 accumulation on specific normalization equations—especially under fp8 training—remains underexplored.

### Clarity
- The discussion around fp32 accumulation flags its importance but omits which exact equations need 32-bit restoration. A detailed breakdown would help readers reproduce results.
- Overall writing can be improved a lot.

### Significance
- Even though the channel level is a new axis, but the author lacks more comprehensive analysis (e.g., each channel's / head sensitivity to the all-reduce). There is no reason we conduct all-reduce for continuous dimension.
- Evaluation is limited to zero-shot accuracy and validation loss on 1 B/7 B models. Extending experiments to larger models (e.g.\ Llama 3.3, Llama 4, Qwen 3) and to NVIDIA GPUs with NVLink would strengthen real-world applicability.

### Originality
- The related-work comparison lacks a head-to-head ablation against techniques like Top-K activation compression, Domino fusion and Ladder Residual.

### Generalizability
- Inference-time tensor-parallel dimension mismatches (§3.3) pose deployment challenges. Empirical results quantifying accuracy vs. synchronization fraction \(p\) during serving would clarify practical trade-offs.

---

> ### Author Rebuttal · Authors · 2025-07-30
>
> We thank the reviewer for the thoughtful and detailed feedback. We are pleased that the reviewer recognizes the novelty and innovation of channel-level synchronization, as well as the importance of the numerical stability considerations we addressed. In response to the reviewer's valuable suggestions, we have expanded our empirical evaluation. These new experiments provide a more comprehensive characterization of CAAT-Net's performance and robustness across diverse workloads. Additionally, we address the reviewer's concerns, particularly those related to partial reduction on contiguous channels, further clarifications on numerical stability, and limitations at inference time.
>
> ---
>
> **(1)**
>
> **Contiguous channel selection**
>
> We thank the reviewer for these important questions about channel selection strategy. There are two compelling reasons for contiguous channel selection:
>
> **Theoretical equivalence**:
> The reduced channels are selected at initialization. Since channels are randomly initialized, any subset of channels is equivalent.
> Specifically, we could permute all weight matrices to make any arbitrary channel subset contiguous, yielding a statistically equivalent initialization. Therefore, contiguous selection imposes no fundamental limitation.
>
> **Implementation efficiency**:
> In our work we aim to use existing communication collectives efficiently. Collectives operate most efficiently on contiguous memory
> regions. Interleaved selection would require either additional operations before communication or custom communication kernels.
>
> Given this theoretical equivalence and practical efficiency advantage, we focused on contiguous selection.
>
> ---
> **(2)**
>
> **Numerical stability**
>
>  We thank the reviewer for seeking clarification on the fp32 accumulation requirements.
>
> The specific operation requiring fp32 accumulation is the gradient summation across devices in the backward pass, corresponding to the left-hand side of Equation 15:
> $$\sum_m \left( \frac{\partial \mathcal{L}(X_m)}{\partial X_m} \cdot \frac{\partial X}{\partial Z_1} \right)$$
>
> This summation represents the all-reduce operation that aggregates gradients from all tensor-parallel devices after they have
> back-propagated through the normalization layer. Importantly, the communication itself can be in 16bit precision, but the all-reduce summation after the communication must be in 32bit.
>
> The normalization function itself remains unchanged. We mentioned normalization operations only as an analogy for other computations that typically require higher precision during training.
>
> Regarding fp8 training: while we have not extensively tested CAAT-Net under fp8, the same principle would likely apply. The gradient
> reduction step would require higher precision to maintain numerical stability.
>
> ---
>
> **(3)**
>
> **Figure redundancy**
>
> We thank the reviewer for the valuable feedback. Figure 1 visualizes the modified two layer MLP architecture at a high level, while Figure 2 provides a detailed explanation of the partial channel-reduce implementation and the corresponding backward pass adjustments. We appreciate the reviewer's suggestion and will consider merging these figures to improve clarity and flow in the
> camera-ready version.
>
> ---
> **(4)**
>
> **Why does synchronizing only a subset of channels preserves correctness.**
>
> We thank the reviewer for this fundamental question about the correctness of partial synchronization.
>
> CAAT-Net does not preserve correctness in the sense of approximating the original fully-synchronized model. Instead, it defines a new architecture specifically tailored for training on multiple devices using partial synchronization. As in all papers which introduce novel architectures, we validate our claims empirically. Our empirical results demonstrate that this specific architectural change preserves learning effectiveness.
>
> ---
>
> **(5)**
>
> **Evaluation of larger models and using Nvidia GPU's**
>
> **Speedup**
>
> Please find speedup for 34B and 70B models in Appendix D (in the supplementary materials). On Llama-70B, our method delivers a 13% speedup at $p = 0.5$ and a 23% speedup at $p = 0.25$, highlighting its continued effectiveness at larger model scales.
>
> Regarding Nvidia GPU's, we conducted additional experiments to show real-world applicability. Due to character limits in our response, we refer the reviewer to section 3 of our rebuttal to 8fqL for full experiment setup and results. We observe up to a 9% speedup for $p = 0.5$ and a 14% speedup for $p = 0.25$ on NVIDIA hardware. Importantly, these results are preliminary and not fully optimized, and we believe they can be further improved.
>
> **Accuracy**
>
> Precisely evaluating accuracy/optimal value of $p$ for larger models requires full training. Most of the LLM training literature experiments with smaller scale models, because of the high compute cost. Even training a 7B model requires a large cluster and is very costly. We show in our paper that our method scales well from 130M to 7B parameters, and expect these trends to continue. For example, for $p=0.5$, we found that accuracy was slightly improved with respect to the baseline for both the
> 130M model and the 7B model. In the regime that we investigated, we did not find evidence that CAAT-Net accuracy deteriorates with increasing model size for a given $p$.
>
> ---
>
> **(6)**
>
> **Scaling cost projections**
>
> In general, the utility of our method depends on many factors, such as hardware, architecture, sequence length, tensor-parallel dimension, etc. Please see Appendix C in the supplementary materials for a detailed speedup analysis, where we model our methods utility as a function of architecture, hardware and tensor-parallel rank.
>
> ---
>
> **(7)**
>
> **Comparison to Top-K, Domino and Ladder Residual**
>
> We thank the reviewer for this question, and use this opportunity to highlight CAAT-Net's advantages compared to other
> methods.
>
> **Comparison to pipelining methods**
>
> There is an important distinction between Ladder-residual and Domino to our method. Ladder-residual and Domino both aim to move communication off the critical path. In practice, both of these approaches do not reduce bandwidth at all. They just enable pipelining between communication and computation. Our approach reduces communication bandwidth.
>
> For specific hardware (with fast interconnect) and for low tensor-parallel dimensions, pipelining can be sufficient. However, this
> approach is not scalable. When increasing the tensor-parallel dimension, computation per device decreases, while communication does not. Total communication time can quickly overcome total computation time. For example, in our paper we show inference speedup for Llama2-7B with tensor-parallel 16. There, setting $p=0$ gives over 50% speedup, so more than half of the time is spent on communication. For larger tensor-parallel dimensions this problem is even more pronounced. For this reason, we do not view CAAT-Net as a competing approach to pipelining optimization, but a complementary one.
>
> Regarding accuracy, for large enough $p$ our method preserves and can even improve accuracy in some cases, as we show in the paper.
> Domino does not affect accuracy as it does not change the underlying transformer architecture.
> The Ladder-Residual paper reports similar accuracy for pretraining a 1.2B parameter model, and measures degradation when training a 3.5B parameter model. These results are based on Table 3 of their paper.
>
> **Comparison to compression methods**
>
> Accuracy: Unlike Ladder-residual and Domino, Top-K reduces communication bandwidth. Results for Top-K compression are available in Appendix E (supplementary materials). CAAT-Net significantly outperform Top-K compression.
>
> Speedup: In general, computing Top-K is very slow and its use overshadows communication benefits. As an example, we present throughput results for the experiments described in Appendix E. Due to character limits in our response, we include the experiment setup and results to section 5 of our response to reviewer YSCj.
>
> ---
>
> **(8)**
>
> **Inference Flexibility**
>
> Inference can run on different device counts than training without finetuning, but with important limitations and trade-offs.
>
> The most straightforward approach is using logical devices. A single physical device can simulate multiple tensor-parallel ranks. For
> example, if trained with TP=2, inference on TP=1 requires the single device to perform both sets of computations sequentially, replacing the partial channel-reduce with local summation. This can increase computation time but preserves model correctness.
>
> Changing the synchronization factor $p$ at inference time is not supported, as this effectively changes the model architecture. A model trained with $p=0.5$ expects specific private and shared channel configurations. Altering $p$ is equivalent to serving a completely different model.
>
> While we have not conducted extensive empirical studies on inference flexibility, our current results suggest the primary deployment path is maintaining the training tensor-parallel configuration for optimal performance. CAAT-Net represents a model tailored to a specific parallelization scheme, whereas traditional approaches separate model architecture from parallelization strategy. While this compromises flexibility, we argue that in an age of increasing model sizes, this is a relevant approach to achieve effective scaling. This is similar to Quantization Aware Training (QAT), in which a model is trained to be served at a specific precision, and is not meant to be used in different precisions [B]. However, these deployment limitations can be addressed through logical devices or fine-tuning approaches to adjust $p$.
>
> ---
>
> [A] PyTorch Labs. gpt-fast, 2024. URL https://github.
> com/pytorch-labs/gpt-fast. Accessed: 2024- 09-29.
>
> [B] K. Du, Y. Zhang, and H. Guan, "From quantized dnns to quantizable
> dnns," CoRR, vol. abs/2004.05284,2020. [Online]. Available:
> https://arxiv.org/abs/2004.05284

---

> > ### Comment · Reviewer_f4Cy · 2025-08-02
> >
> > Thanks for answering my question.
> >
> > > Given this theoretical equivalence and practical efficiency advantage, we focused on contiguous selection.
> >
> > I am still confused here. For example, in a lot of LLM papers, such as *DuoAttention: Efficient Long-Context LLM Inference with Retrieval and Streaming Heads* some head is more important than others. There is 'NO Equivalent' for channel. Regarding the premitives of communication, I think we can do some basic ablation here.
> >
> > > Evaluation of larger models and using Nvidia GPU's
> >
> > I see the results of appendix now. Thanks for pointing this out. I still have these questions:
> >
> > * Could you specify exactly which communication library and configuration you used (e.g. NCCL, MPI, custom RPC), including collective operations, network topology, buffer sizes, and any tuning parameters? This will help others reproduce your results.
> > * In your experiments, the measured speedup for $P=0.75$ is very small (which is most lossy way for the model other p is too lossy)—indeed, sometimes worse than the unmodified baseline. I think contribution of this method is too weak then.
> > * You reduce bandwidth by dropping tails, but it appears the accuracy or end-to-end throughput gains may not justify the partial-communication overhead. Could you share any profiling or empirical analysis that quantifies this cost/benefit trade-off?
> >
> > > between Ladder-residual and Domino to our method. Ladder-residual and Domino both aim to move communication off the critical path. In practice, both of these approaches do not reduce bandwidth at all. They just enable pipelining between communication and computation. Our approach reduces communication bandwidth.
> >
> > This is exact my point, Ladder-Residual and Domino fully overlap communication and computation, yet they don’t lower bandwidth. If communication can be perfectly pipelined off the critical path, why is further bandwidth reduction still necessary? What scenarios break overlap and make bandwidth savings crucial?
> >
> > > Inference Flexibility
> >
> > Unlike fixed-bit quantization, TP-size (tensor parallelism width) can vary dynamically at inference time. What are the practical trade-offs between switching communication patterns on the fly versus sticking with a single strategy? Logical devices is a smart way (like how does PD-disaggregate work), but how do we trade-off we can do tp=1 essentally but we sacrifice accuracy to tp=2 logically for actual tp=1?

---

> ### Author Response · Authors · 2025-08-04
>
> We thank the reviewer for reading our response in detail. Below we address the remaining comments.
>
> ---
>
> **(1)**
>
> **Contiguous reduction and DuoAttention**
>
> We fully agree that head differentiation develops throughout training. However, if we choose the heads before we train, and even before we sample the random initialization, then there is provably no difference between heads. For example, selecting a specific seed at initialization might lead to the head on device number 1 to be a retrieval head, and selecting a different seed might lead the head on device number 2 to be a retrieval head. The same applies to channels. In the networks we train, there is no significance to specific channels before training (except for in RoPE, which treats channels differently, but this doesn't affect partial reduction). Their significance is only given after we start training, i.e after we select the contiguous chunk.
>
> As per the reviewer's request, we ran the necessary ablations. Regarding speedup, to avoid implementing a new collective, we copied the random channels that are reduced into a new, contiguous tensor, and reduced that tensor. In our initial implementation, this is over 10x slower than vanilla transformers.
>
> Regarding accuracy, we ran the 130M model with $p=0.5$ over 2.6B tokens with 4 different seeds which select different random channels to be reduced. We measure no degradation for contiguous reduction. See results below:
>
> | **Configuration** | **Validation loss** |
> |-------------------|---------------------|
> | Contiguous        | 2.02                |
> | Random seed 1          | 2.02                |
> | Random seed 2          | 2.02                |
> | Random seed 3          | 2.04                |
> | Random seed 4          | 2.04                |
>
> ---
>
> **(2)**
>
> **Communication configuration**
>
> For Gaudi experiments we use the HCCL library which is an Intel collective library. Communication collectives are implemented with torch.distributed.all_reduce(). Each pair of Gaudi3 devices has a 75GB/s connection, so in total each Gaudi has 525GB/s intra-node connection. Furthermore, each Gaudi3 device has a 75GB/s inter-node connection.
>
> The Nvidia experiments (detailed in section 3 of our rebuttal to 8fqL) uses the NCCL backend.  All of the communication collectives are implemented with torch.distributed.all_reduce(). We used NCCL's default buffer sizes and did not set any custom variables. All collectives leverage intra-node NVLink connectivity.
>
> ---
>
> **(3)**
>
>
> **$p=0.75$ is too lossy**
>
> We disagree with the statement that the model is too lossy for values of $p$ lower that 0.75. In the paper we show no accuracy degradation for $p=0.5$. Furthermore, please see section 3 of our response to reviewer 7RqG, where we show good accuracy results up to $p=0.3$ for the 1B TinyLlama model.
>
> ---
>
> **(4)**
>
> **Profiling**
>
> Unfortunately due to guidelines we cannot share profiling results. However, in our research we find that using a certain value of $p$ reduces communication time by a factor of approximately $p$. For example, see all-reduce times for a Llama2-7B network trained on Intel Gaudi3 with micro-batch size 1 and tensor-parallel 8:
>
> | **p = 1** | **p = 0.75** | **p = 0.5** | **p = 0.25** |
> |-----------|--------------|-------------|--------------|
> | 870 μs    | 650 μs       | 460 μs      | 240 μs       |
>
> ---
>
> **(5)**
>
> **Comparing to pipelining methods**
>
> Communication and computation cannot be pipelined in the following scenarios:
> - Total all-reduce communication time overcomes total computation time at high tensor-parallel. For example, suppose we use Llama2-7b with batch size 1 and tensor-parallel 32. This can used to achieve extremely low latency inference. In the prefill stage, on Intel Gaudi3, we measure the computation time per block to be around 650 $\mu s$ while communication is around 850 $\mu s$, making it impossible to completely pipeline communication and computation.
> - Other parallelism types interfere with TP-related communication.  One of many examples is context-parallelism, where communication time alone can exceed compute time (see Tables 4 and 7 in [A]). When adding TP after attention blocks that use context-parallelism, there’s no compute left to pipeline both types of communication.
> ---
>
> **(6)**
>
> **Logical devices**
>
> Regarding inference with varying tensor-parallel dimensions, we do not expect the challenge to be different with CAAT-Net than it is in other situations. The important thing to note is that when doing inference using logical devices, there is a chance of speedup degradation (depending on workload, environment, etc.).
> If we understand the reviewer's question correctly, then yes, we will have tp=2 accuracy for actual tp=1. We point out that tp=2 accuracy might be equal to or better than tp=1 accuracy. Please see section 3 of our response to reviewer 7RqG.
>
> ---
>
> [A] Amy Yang et. al. Context parallelism for scalable million-token inference, 2024.
> URL https://arxiv.org/abs/2411.01783.

---

> > ### Author Response · Authors · 2025-08-08
> >
> > As there are fewer than 24 hours remaining in the discussion period, we hope to hear from the reviewer on whether our response has resolved the remaining concerns.

---

> > > ### Comment · Reviewer_f4Cy · 2025-08-08
> > >
> > > The technical concerns I had with this paper are mostly resolved. I will raise my point.
> > >
> > > Please, include our discussion in the full paper/appendix and include the points below.
> > >
> > > 1) Ladder-residual is also trained from scratch. I hope to see at least a discussion in the related work or a comprehensive analysis in the appendix.
> > > 2) I am still doubting *‘we show no accuracy degradation’*, please include more accuracy training results for larger models.
> > >
> > > Thanks.

---

> > > > ### Author Response · Authors · 2025-08-08
> > > >
> > > > We thank the reviewer for the valuable suggestions. We will include this discussion in the paper and address the points the reviewer raised.

---

### Official Review · Reviewer_YSCj · 2025-07-05

**Clarity:** 2
**Significance:** 3
**Originality:** 3
**Rating:** 4
**Confidence:** 4

**Summary:**

This paper introduces CAAT-Net (Communication-Aware Architecture for Tensor-Parallelism), a new method for reducing communication overhead in tensor-parallel training and inference of Large Language Models (LLMs). The key idea is to partially synchronize activation tensors across devices instead of performing full all-reduce operations. The authors propose a new primitive called partial channel–reduce, which synchronizes only a portion of the hidden dimension channels (controlled by parameter p), thereby reducing communication bandwidth by up to 50%.

**Questions:**

Questions：

1.Could the authors provide more theoretical or empirical justification for the √r scaling of private channels at initialization? Would removing this scaling lead to divergence or worse convergence? A controlled ablation would help clarify.

2.Why are direct comparisons to pipelining approaches or Top-K compression baselines not included? The controlled experiment could quantify whether partial synchronization outperforms in realistic scenarios.

3.Have the authors considered evaluating CAAT-Net on other open-source architectures? This would help demonstrate whether the proposed partial synchronization mechanism generalizes beyond the LLaMA-based models currently used in the experiments. Adding such results could strengthen the empirical validation.

4.While Section 5.2 explores different p values on a 130M model, the main large-scale experiments (1B and 7B) fix p = 0.5 without further justification. It remains unclear whether p = 0.5 is optimal or just a safe default. A more systematic ablation of p on larger models would help clarify the tradeoff between communication cost and accuracy.

**Ethical Concerns:**

["NO or VERY MINOR ethics concerns only"]

**Final Justification:**

I have read the authors' response. Thanks for the rebuttal. The response mainly addressed my concerns. Considering other reviewers' comments and rating scores, I will raise my rating to 4 accordingly.

**Limitations:**

None.

**Paper Formatting Concerns:**

None.

**Quality:**

2

**Strengths And Weaknesses:**

Strengths：

1.The paper proposes CAAT-Net that applies partial channel–reduce to reduce synchronization overhead. CAAT-Net lowers communication volume during tensor-parallel training and inference.

2.The paper describes modifications to the backward pass required by partial synchronization, including changing the placement of the all-reduce operation and using fp32 accumulation to mitigate numerical differences.

3.The paper provides a design of the partial channel–reduce mechanism, including its integration into both MLP and attention layers.

Weaknesses：

1.In section 3.2, the variance scaling adjustment in partial synchronization is briefly mentioned, but lacks deeper theoretical justification or ablation to validate its necessity and effect.

2.The experimental evaluation focuses on a limited set of model scales (130M, 1B, 7B) and primarily on the LLaMA architecture. Including results on other widely used open-source models and stronger baselines would further strengthen the empirical claims.

3.In the partial channel–reduce mechanism, only a subset of channels is synchronized across devices. It is not clearly discussed how the private (unsynchronized) activations contribute to downstream layers.

4.The paper frames CAAT-Net as an architectural design, but the actual modification primarily involves a replacement of all-reduce with partial-reduce and a slight adjustment to initialization. It is unclear to what extent this constitutes a new “architecture” versus a communication primitive. Clarifying this distinction and justifying the architectural framing would improve clarity.

---

> ### Author Rebuttal · Authors · 2025-07-31
>
> We thank the reviewer for the constructive and thoughtful feedback. We appreciate the recognition of our contributions to reducing communication overhead and the required changes to the backward pass. We respond to the reviewers comments and questions below. Furthermore, we have expanded our experiments to include more diverse models and broader ablations over $p$.
>
> ---
>
> **(1)**
>
> **Questions**
>
> The paper frames CAAT-Net as an architectural design, but the actual modification primarily involves a replacement of all-reduce with partial-reduce and a slight adjustment to initialization. It is unclear to what extent this constitutes a new “architecture” versus a communication primitive. Clarifying this distinction and justifying the architectural framing would improve clarity.
>
> In the partial channel–reduce mechanism, only a subset of channels is synchronized across devices. It is not clearly discussed how the private (unsynchronized) activations contribute to downstream layers.
>
> **Answer**
> We acknowledge that the core innovation involves modifying the communication primitive. While this seems like a slight adjustment, in practice it constitutes an architectural change. Partial channel–reduce fundamentally alters the model's computation graph and information flow. This distinguishes CAAT-Net from communication optimizations like pipelining or compression, which preserve or approximate the underlying mathematical operations. For example, the MLP in CAAT-Net can no longer be expressed as two standard linear transformations with a non-linearity between them, as shown in Section 3. Instead, we need to redefine both the MLP and attention mechanisms to handle device-specific inputs and outputs.
>
> Moreover, supporting this design requires non-trivial modifications beyond the communication operation and initialization. These modifications are detailed in Section 4.
>
> Regarding the private (unsynchronized) activations: they are not discarded or approximated. They flow through the same downstream layers as shared activations, contributing fully to the model’s expressivity.
>
> ---
>
>
> **(2)**
>
> **Question**
> In section 3.2, the variance scaling adjustment in partial synchronization is briefly mentioned, but lacks deeper theoretical justification or ablation to validate its necessity and effect.
>
> **Answer**
> We thank the reviewer for this question. The variance scaling adjustment addresses a statistical property of partial channel-reduce that we can clarify with a more detailed theoretical analysis.
>
> **Theoretical Justification:**
>
> Consider the activations in $Z_1$ after partial channel-reduce. Assuming the MLP outputs before reduction ($Y_1B_{11}$, $Y_2B_{12}$, $Y_1B_{21}$) have zero mean and variance $\sigma^2_A$ and are independent:
>
> For shared channels:
>
> \begin{equation}
>     \text{Var}(Y_1B_{11} + Y_2B_{12}) = \text{Var}(Y_1B_{11}) + \text{Var}(Y_2B_{12}) = 2\sigma^2_A
> \end{equation}
>
> For private channels:
>
> \begin{equation}
>     \text{Var}(Y_1B_{21}) = \sigma^2_A
> \end{equation}
>
> This variance mismatch can lead to uneven gradient signals across channels. By scaling private channel weights by $\sqrt{2}$ at initialization, we achieve uniform variance of $2\sigma^2_A$ across all channels.
>
> **Empirical Validation:**
>
> We validated this approach on 130M parameter models ($p=0.5$, 5B tokens), observing the following improvements:
>
> |                        | TP=16 | TP=8  |
> | ---------------------- | ----- | ----- |
> | Regular Initialization | 1.94  | 1.927 |
> | Altered Initialization | 1.926 | 1.924 |
>
> The improvements are modest but support the theoretical motivation.
>
> **Response to Reviewer Concern:**
>
> While this initialization provides theoretical and empirical benefits, it represents a secondary contribution compared to the core architectural changes in Section 3 and 4. We propose moving this discussion to the appendix with expanded ablation studies for the camera-ready version, allowing the main text to focus on the essential modifications.
>
> ---
>
> **(3)**
>
> **Question**
>
> The experimental evaluation focuses on a limited set of model scales (130M, 1B, 7B) and primarily on the LLaMA architecture. Including results on other widely used open-source models and stronger baselines would further strengthen the empirical claims.
>
> **Answer**
>
> Different architecture:
>
> To address this concern, we extended our experimentation to GPT3-XL [A], a 1.3B parameter model. We find that CAAT-Net maintains and even slightly improves model accuracy. Due to character limitation, we refer the reviewer to the section 3 of our rebuttal to reviewer 7RQG to see the full experiments and data.
>
> Model scale:
>
> If the concern pertains to speedup measurements on larger models, we refer the reviewer to Appendix D.1, where we report inference speedups on larger LLaMA models, including LLaMA-34B and LLaMA-70B. On LLaMA-70B, our method achieves a 13\% speedup at $p=0.5$ and 23\% at $p=0.25$, demonstrating the effectiveness of our approach at scale.
>
> If the concern instead relates to training-time accuracy at such scales, we note that most prior work is evaluated on smaller models (130M–7B parameters), given the immense cost of training. For instance, training our 7B model already requires substantial compute resources and access to a large cluster, which is beyond the capabilities of many academic groups. Notably, recent prominent works like GaLore [B] and Rho1 [C] also focus on models up to 7B parameters.
>
> Our results demonstrate that the method scales effectively from 130M to 7B parameters, consistently preserving accuracy while improving performance.
>
> ---
>
> **(4)**
>
> **Question**
>
> While Section 5.2 explores different p values on a 130M model, the main large-scale experiments (1B and 7B) fix p = 0.5 without further justification. It remains unclear whether p = 0.5 is optimal or just a safe default.
>
> **Answer**
>
> $p=0.5$ was indeed chosen as a safe default for the large experiments. To further study the effects of tensor-parallel dimension and $p$, we added many data points to Table 2. We show that our method performs well on small models when $p$ is sufficiently large, sometimes even yielding slight improvements in validation loss. Furthermore,  we include additional experiments on TinyLlama with varying values of $p$, demonstrating that these trends scale consistently. Due to character limitation, we refer the reviewer to the section 3 of our rebuttal to reviewer 7RQG to see the full experiment setup and data.
>
> ---
>
> **(5)**
>
> **Question**
>
> Why are direct comparisons to pipelining approaches or Top-K compression baselines not included? The controlled experiment could quantify whether partial synchronization outperforms in realistic scenarios.
>
> **Answer**
>
> Comparison to pipelining approaches:
>
> CAAT-Net differs fundamentally from pipelining methods such as Domino. While pipelining methods aim to hide communication latency by overlapping it with computation, CAAT-Net directly reduces the amount of data communicated.
>
> Pipelining can be effective on low tensor-parallel dimensions with fast interconnects, but this approach does not scale well. As tensor-parallelism increases, the computation per device decreases, while the communication doesn't. Consequently, communication can dominate total runtime. For example, in our Llama2-7B experiment with tensor-parallelism of 16, setting $p=0$ achieves over 50\% inference speedup. This implies that more than half of the workload time is spent on communication. This imbalance becomes even more pronounced at larger tensor-parallel scales. For this reason, we view CAAT-Net as complementary—not competing—to pipelining methods.
>
> In terms of accuracy, CAAT-Net preserves and in some cases improves accuracy, particularly at larger values of $p$, as shown in the paper. Domino preserves accuracy because it does not alter the transformer architecture.
>
> Comparison to Compression Methods:
>
> Accuracy: Unlike pipelining approaches, Top-K compression does reduce communication bandwidth. However, as shown in Appendix E, CAAT-Net achieves significantly better accuracy than Top-K compression.
>
> Speedup: In general, computing Top-K is very slow and its use overshadows communication benefits. As an example, we present throughput results in TFLOP/s for the experiments on the 130M model described in Appendix E. To be consistent with CAAT-Net notation, $p$ is the percentage of the original tensor communicated (so $p=0$ is no communication and $p=1$ is full communication). To calculate the Top-K we use torch.topk(). Using the Top-K operation completely degrades throughput. Although this experiment is on a small model, this severe performance degradation is evident and holds for larger models as well. We can add more experiments on larger models for the camera-ready version if the reviewer thinks it is necessary.
>
> | \$p\$ | Throughput (TFLOP/s) |
> | ----- | -------------------- |
> | 1     | 101.1                |
> | 0.9   | 2.9                  |
> | 0.7   | 3.8                  |
> | 0.5   | 5.6                  |
> | 0.3   | 9.8                  |
> | 0.1   | 24.7                 |
>
> ---
>
> [A] Tom Brown, Benjamin Mann, Nick Ryder, Melanie Subbiah, Jared D Kaplan, Prafulla Dhariwal,
> Arvind Neelakantan, Pranav Shyam, Girish Sastry, Amanda Askell, et al. Language models are
> few-shot learners. Advances in neural information processing systems, 33:1877–1901, 2020.
>
> [B] J. Zhao, Z. Zhang, B. Chen, Z. Wang, A. Anandkumar, and Y. Tian,
> “Galore: Memory-efficient llm training by gradient low-rank projection,” arXiv preprint arXiv:2403.03507, 2024.
>
> [C] Zhenghao Lin, Zhibin Gou, Yeyun Gong, Xiao Liu, Yelong Shen, Ruochen Xu, Chen Lin, Yujiu Yang, Jian
> Jiao, Nan Duan, et al. 2024b. Rho-1: Not all tokens
> are what you need. arXiv preprint arXiv:2404.07965.

---

> ### Author Response · Authors · 2025-08-06
>
> As the end of the discussion period is soon approaching, could the reviewer kindly let us know if there are any remaining concerns we should address? We have responded to the comments and questions raised by the reviewer and would appreciate the chance to address any further feedback.

---

### Decision · Program_Chairs · 2025-09-17

**Decision:**

Accept (poster)

**Comment:**

This paper addresses the high communication cost of tensor-parallelism in training LLMs by introducing minor adjustments that reduce communication by 50% without sacrificing accuracy, validated on 1B and 7B models. The approach is simple yet effective, accelerating both training and inference. While the method lacks theoretical backing, the experimental setup is reasonable and the algorithm itself is sound.

Concerns about the scale and model size are not critical here: reducing tensor-parallel communication is an important challenge, and this work makes a valuable contribution toward addressing it. Although it is not systematically proven that all transformers would benefit, the paper’s ablations help clarify the method’s limitations. Overall, the contribution is novel, meaningful and timely, and the rebuttal adequately addressed the main weaknesses.